# Epigenome-wide association meta-analysis of DNA methylation with coffee and tea consumption

Irma Karabegović[1,2,3], Eliana Portilla-Fernandez[1], Yang Li[4], Jiantao Ma[5,6], Silvana C. E. Maas[1,2], Daokun Sun[7], Emily A. Hu[8], Brigitte Kühnel[9,10], Yan Zhang[11], Srikant Ambatipudi[12,13,14], Giovanni Fiorito[15,16], Jian Huang[16,17,18], Juan E. Castillo-Fernandez[19,20], Kerri L. Wiggins[21], Niek de Klein[22], Sara Grioni[23], Brenton R. Swenson[21], Silvia Polidoro[24,16], Jorien L. Treur[25], Cyrille Cuenin[14], Pei-Chien Tsai[19,26,27], Ricardo Costeira[19], Veronique Chajes[28], Kim Braun[1], Niek Verweij[22,29], Anja Kretschmer[9,10], Lude Franke[22,30], Joyce B. J. van Meurs[31], André G. Uitterlinden[31], Robert J. de Knegt[4], M. Arfan Ikram[1], Abbas Dehghan[16,17], Annette Peters[9,10,32], Ben Schöttker[11], Sina A. Gharib[33], Nona Sotoodehnia[21], Jordana T. Bell[19], Paul Elliott[16,17,18,34], Paolo Vineis[16], Caroline Relton[12], Zdenko Herceg[14], Hermann Brenner[11,35,36,37], Melanie Waldenberger[9,10,32], Casey M. Rebholz[8], Trudy Voortman[1], Qiuwei Pan[4], Myriam Fornage[7], Daniel Levy[6], Manfred Kayser[2] & Mohsen Ghanbari[1,38✉]

Coffee and tea are extensively consumed beverages worldwide which have received considerable attention regarding health. Intake of these beverages is consistently linked to, among others, reduced risk of diabetes and liver diseases; however, the mechanisms of action remain elusive. Epigenetics is suggested as a mechanism mediating the effects of dietary and lifestyle factors on disease onset. Here we report the results from epigenome-wide association studies (EWAS) on coffee and tea consumption in 15,789 participants of European and African-American ancestries from 15 cohorts. EWAS meta-analysis of coffee consumption reveals 11 CpGs surpassing the epigenome-wide significance threshold (P-value $<1.1\times10^{-7}$), which annotated to the *AHRR*, *F2RL3*, *FLJ43663*, *HDAC4*, *GFI1* and *PHGDH* genes. Among them, cg14476101 is significantly associated with expression of the *PHGDH* and risk of fatty liver disease. Knockdown of *PHGDH* expression in liver cells shows a correlation with expression levels of genes associated with circulating lipids, suggesting a role of *PHGDH* in hepatic-lipid metabolism. EWAS meta-analysis on tea consumption reveals no significant association, only two CpGs annotated to *CACNA1A* and *PRDM16* genes show suggestive association (P-value $<5.0\times10^{-6}$). These findings indicate that coffee-associated changes in DNA methylation levels may explain the mechanism of action of coffee consumption in conferring risk of diseases.

A full list of author affiliations appears at the end of the paper.

Excluding water, coffee and tea are the most commonly consumed beverages around the world. The preference for one or the other and the quantity of coffee and tea consumed vary between individuals[1], which can be influenced by the geographical region as well as cultural and personal preference. In addition, both coffee and tea are sources of complex compounds with different chemical classes, the most commonly known is caffeine[2]. Caffeine belongs to the methylxanthines family, which consists of frequently ingested pharmacologically active substances, for example, through the stimulation of the central nervous system[3]. Although caffeine is present in both coffee and tea, its concentration in tea is much lower[4]. Moreover, both beverages differ on the bioavailability of polyphenols and other chemical compounds[5]. The biochemistry of coffee and tea has been extensively documented, indicating that different roasting, temperatures, or brewing of the two can impact the abundancy and bioavailability of their complex compounds[6,7]. Moreover, it has been shown that the lipid content of boiled un-filtered coffee may be as much as 60 times higher than the lipid content of filtered coffee[8]. There has been an ongoing debate as to whether habitual consumption of coffee[9] and tea[10] is beneficial or harmful to health. The conclusion varies among outcomes; for example: lowering the risk for type 2 diabetes (T2D), cardiovascular diseases[11,12], liver diseases[13,14], and overall mortality[15]; or increasing serum levels of low-density lipoprotein (LDL) and total cholesterol[16]. These contradictory findings in observational studies seem to be determined by the presence of different compounds in the two beverages, especially regarding coffee consumption[9]. Yet, the biological mechanisms underlying associations of coffee and tea consumption with risk of diseases remain to be understood.

Epigenetics represents modifications to DNA that do not change the underlying DNA sequence, but can influence gene expression[17]. The most extensively studied epigenetic mechanism so far is DNA methylation, where a methyl group ($-CH_3$) is added to or removed from the cytosine nucleotide that is followed by a guanine nucleotide in the DNA sequence, known as Cytosine-Phosphate Guanine (CpG) site[17], resulting in altered gene expression. The DNA methylation levels differ by age, sex and lifestyle factors, including dietary exposures[18–20]. Here we postulated that alteration of DNA methylation via coffee or tea consumption is an underlying mechanism linking the intake of these beverages to health outcomes. Previous epigenome-wide association studies (EWAS) have reported suggestive association of some CpGs with tea or coffee consumption[21,22]; however, these studies were limited by the modest sample sizes. In this work, we conduct large-scale EWAS meta-analyses of coffee as well as tea consumption in 15,789 participants of European and African–American ancestries from 15 cohort studies. For the associated CpGs, we evaluate their correlations with genetic variation and gene expression. Additionally, we seek to identify the potential causal effect of coffee consumption on the associated CpGs and different health outcomes. Finally, we perform experimental studies for a gene annotated to a coffee-associated CpG to investigate its link to lipid metabolism and liver diseases.

## Results

Figure 1 depicts an overview of the study flow. Characteristics of the cohorts participating in the discovery ($n = 9612$) and replication phase ($n = 6177$) are presented in Table 1. The mean age across all participating cohorts ranged from 41.1 years in the Airwave cohort to 78.6 years in the CHS_EA cohort. The majority of the study participants were women (61.44%). Mean total coffee intake among cohorts ranged from 0.6 cups/day in the CHS_AA cohort to 3.5 cups/day in the RS-III-2 cohort, while mean total tea

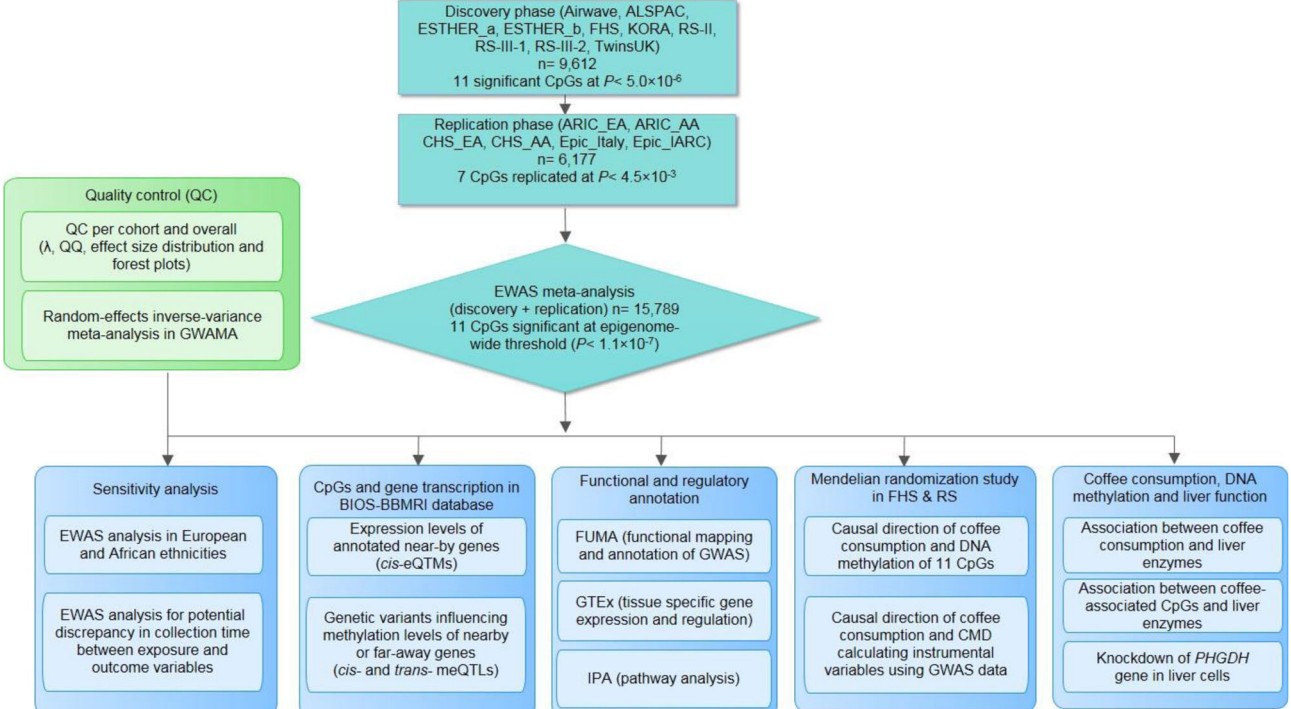

**Fig. 1 Overview of the study flow.** The flowchart summarizes our study design including EWAS meta-analysis to identify DNA methylation sites associated with coffee and tea consumption, and post-EWAS in silico and in vitro experiments. QQ quantile–quantile, eQTM *cis*-expression quantitative trait methylation, meQTL methylation quantitative trait loci, RS Rotterdam Study, FHS Framingham Heart Study, ALSPAC The Avon Longitudinal Study of Parents and Children, CHS Cardiovascular Health Study, ARIC The Atherosclerosis Risk in Communities Study, EPIC Prospective Investigation into Cancer and Nutrition, KORA Cooperative Health Research in the Augsburg Region Study.

**Table 1 Characteristics of the cohort participants.**

| Cohort | N | ET | Age (years) | Women (%) | Coffee (cups/day) | Tea (cups/day) | Smoking Current | Former | Never | BMI (kg/m²) | Alcohol (g/day) |
|---|---|---|---|---|---|---|---|---|---|---|---|
| *Discovery phase* | | | | | | | | | | | |
| RS-III-2 | 149 | EA | 61.01 (4.42) | 116 (66.3%) | 3.52 (2.22) | 1.43 (1.5) | 20 (11.4%) | 89 (50.9%) | 66 (37.7%) | 27.47 (4.13) | 8.18 (7) |
| RS-II-3 | 367 | EA | 70.97 (3.21) | 242 (55.09%) | 2.85 (1.78) | 1.18 (1.3) | 38 (8.8%) | 243 (56.1%) | 152 (35.1%) | 27.69 (4.10) | 8.55 (7.55) |
| RS-III-1 | 549 | EA | 58.89 (7.8) | 283 (51.5%) | 3.49 (2.13) | 1.23 (1.3) | 148 (27%) | 235 (42.8%) | 166 (30.2%) | 27.27 (4.65) | 9.14 (8.68) |
| ALSPAC | 331 | EA | 48.04 (4.03) | 331 (100%) | 1.69 (1.65) | 3.02 (2.08) | Current + former = 152 (46%) | | 179 (54%) | 25.67 (4.67) | 6.5 (6.66) |
| KORA | 1535 | EA | 54 (8.8) | 774 (50.4%) | 3.4 (3) | 1.1 (2.1) | 307 (20%) | 568 (37%) | 660 (43%) | 27.7 (4.48) | 17.3 (22.87) |
| FHS | 3718 | EA | 59 (13) | 2008 (54%) | 1.35 (1.28) | 0.37 (0.77) | 297 (8%) | Never + former = 3421 (92%) | | 28 (5.40) | 10.80 (15.50) |
| ESTHER_A | 973 | EA | 62 (6.5) | 488 (50.1%) | 1.7 (1.1) | 0.4 (0.7) | 315 (32.8%) | 464 (48.3%) | 315 (32.8%) | 27.8 (4.2) | 12.3 (15.4) |
| ESTHER_B | 532 | EA | 62 (6.6) | 327 (61.5%) | 1.7 (1.1) | 0.5 (0.8) | 177 (34.1%) | 426 (47.4%) | 177 (34.1%) | 27.6 (4.40) | 10.9 (14.50) |
| TwinsUK | 552 | EA | 58.45 (9.97) | 552 (61.5%) | 2.37 (2.46) | 3.35 (2.94) | 149 (27%) | 337 (61%) | 149 (27%) | 26.51 (4.94) | 6.37 (9.15) |
| Airwave | 906 | EA | 41.6 (9.3) | 380 (41.9%) | 1.8 (2) | 3.1 (2.6) | 217 (24%) | 596 (65.8%) | 217 (24%) | 27.2 (4.50) | never = 64 (7.1%); current = 777 (85.8%) former = 65 (7.1%) |
| Lifelines | 186 | EA | 45.04 (13.6) | 160 (57%) | 3.94 (2.03) | N/A | 99 (35.2%) | 148 (59.9%) | 99 (35.2%) | 25.78 (4.30) | 8.52 (9.42) |
| *Replication phase* | | | | | | | | | | | |
| ARIC_EA | 1099 | EA | 57.73 (6.02) | 641 (58.33%) | 1.94 (1.99) | 0.77 (1.14) | 232 (21.11%) | 379 (34.49%) | 488 (44.40%) | 26.08 (4.40) | 5.34 (12.07) |
| ARIC_AA | 2736 | AA | 54.11 (6.12) | 1748 (63.89%) | 0.99 (1.33) | 0.38 (0.71) | 760 (27.78%) | 681 (24.89%) | 1295 (47.33%) | 29.89 (6.14) | 4.49 (14.07) |
| CHS_EA | 195 | EA | 78.6 (4.67) | 120 (61.5%) | 0.97 (1.33) | 0.57 (0.97) | 92 (47.2%) | 13 (6.6%) | 85 (43.6%) | 26.8 (4.61) | 2.27 (7.69) |
| CHS_AA | 185 | AA | 75.9 (4.71) | 121 (65.4%) | 0.6 (0.97) | 0.35 (0.62) | 79 (42.7%) | 19 (10.2%) | 80 (43.2%) | 28.3 (4.99) | 1.27 (4.23) |
| EPIC_Italy | 1096 | EA | 53.61 (6.95) | 761 (69%) | 2.79 (1.74) | 0.29 (0.61) | 299 (27%) | 296 (27%) | 501 (46%) | 25.98 (4.07) | 14.5 (18.79) |
| EPIC_IARC | 866 | EA | 52.23 (8.97) | 866 (100%) | 1.25 (1.18) | 0.69 (1.13) | 189 (22%) | 186 (21%) | 491 (57%) | 25.8 (4.45) | 9.1 (12.5) |

*ET Ethnicity, RS Rotterdam Study, ALSPAC The Avon Longitudinal Study of Parents and Children, KORA Cooperative Health Research in the Augsburg Region Study, FHS Framingham Heart Study, ARIC The Atherosclerosis Risk in Communities study, CHS Cardiovascular Health Study, EPIC Prospective Investigation into Cancer and Nutrition, EA European ancestry, AA African ancestry.*

intake ranged from 0.3 cup/day in the EPIC_Italy cohort to 3.4 cups/day in TwinsUK (Table 1).

Quantile–quantile (QQ) plots were generated and the corresponding lambda value computed for the overall meta-analysis of the discovery and replication panels combined, indicated no statistical inflation in the fully adjusted models for coffee or tea consumption (Supplementary Fig. 3). Furthermore, we inspected the effect-size distribution plots indicating that one cohort (Lifelines, $n = 186$) had an effect-size scale non-comparable to other participating cohorts (Supplementary Fig. 4). As the Lifelines cohort had a high standard deviation for coffee consumption, and also no data on tea consumption, we excluded this cohort from further analysis.

EWAS meta-analysis of 9612 participants with European ancestry in the discovery phase identified 11 CpG sites associated with coffee consumption at the suggestive significance threshold of $P < 5.0 \times 10^{-6}$ in Model 2 (Table 2). We sought replication of these CpGs in independent cohorts of both ancestries (EA and AA) comprising 6177 participants, where seven CpGs were successfully replicated with $P < 4.5 \times 10^{-3}$ (0.05/11 CpGs) in the same direction. In the combined meta-analysis of all participants in the discovery and replication cohorts, 11 CpGs passed the epigenome-wide significance threshold ($P < 1.1 \times 10^{-7}$) (Table 2). A Manhattan plot showing the EWAS on coffee consumption is depicted in Fig. 2A. Forest plots for the significantly associated CpGs showed small effects, but an overall consistency in direction across the participating cohorts (Supplementary Fig. 5). Heterogeneity was also assessed; for those CpGs showing a nominal evidence of heterogeneity ($P < 0.05$), we additionally provided results from random-effects inverse-variance meta-analysis (Supplementary Table 4). The CpG with the most significant association with coffee consumption was cg05575921 ($P = 2.17 \times 10^{-15}$, $\beta = -0.0016$) annotated to *AHRR*, a repressor of the *AHR* (Aryl Hydrocarbon Receptor) gene (Fig. 3A). After excluding the largest contributing cohort (FHS) from the meta-analysis of European cohorts, four CpGs namely cg05575921 ($P = 1.2 \times 10^{-14}$, $\beta = -0.003$), cg25648203 ($P = 5.4 \times 10^{-10}$, $\beta = -0.001$), cg21161138 ($P = 5.7 \times 10^{-10}$, $\beta = -0.001$) and cg03636183 ($P = 4.1 \times 10^{-8}$, $\beta = -0.001$) remained significant.

We observed no significant association with tea consumption at the epigenome-wide significance threshold ($P < 1.1 \times 10^{-7}$) even in the EWAS meta-analysis of all participating cohorts ($n = 15,789$). The most significant associations observed for two CpGs cg20099906 (annotated to *CACNA1A*) and cg0584170 (annotated to *PRDM16*) at the borderline threshold of $P < 5.0 \times 10^{-6}$ (Table 3). The Manhattan plot showing the EWAS results on tea consumption is depicted in Fig. 2B.

In order to further examine the novelty of our findings, we investigated whether the loci identified for coffee and tea consumption have been reported previously by GWAS or other EWAS. None of the loci had been associated before with either coffee or tea consumption (Supplementary Table 5). In addition, we looked up the association of *cis*-meQTLs of the coffee-associated CpGs in the publicly available GWAS of coffee intake, published with UK-Biobank data ($n = 358,093$)[23] and available through GWAS ATLAS (https://atlas.ctglab.nl/). We did not find any of the lead SNPs of CpGs-meQTLs to be associated with coffee intake (Supplementary Table 6).

When excluding the cohorts with different time points of collection between methylation and beverage intake data, the results of association between DNA methylation and coffee or tea consumption did not change substantially (Supplementary Table 7). For investigating the potential ancestry effects, we performed an EWAS meta-analysis of coffee consumption separately in EA ($n = 12,868$) and AA ($n = 2921$) participants. Out of the 9 CpGs significantly associated with coffee consumption in

**Table 2 Inverse-variance weighted fixed effects meta-analysis of EWAS with coffee consumption.**

| CpG | CHR | Position | Gene | Discovery phase (n = 9612) | | Replication phase (n = 6177) | | Overall meta-analysis n = 15,789 | | | | |
|---|---|---|---|---|---|---|---|---|---|---|---|---|
| | | | | β (SE) | P-value | β (SE) | P-value | β (SE) | P-value | I² | Direction | Het P-v |
| cg05575921 | 5 | 373378 | AHRR | −0.0014 (2E−04) | 2.33E−10 | −0.0027 (5E−04) | 7.93E−08 | −0.0016 (2E−4) | 2.17E−15 | 70.3 | --------+-----+ | 0 |
| cg25648203 | 5 | 395444 | AHRR | −0.001 (2E−04) | 1.87E−09 | −0.0011 (2E−04) | 8.02E−06 | −0.001 (1E−4) | 7.31E−14 | 21.7 | ---+----+ | 0.2 |
| cg03636183 | 19 | 17000585 | F2RL3 | −0.0014 (2E−04) | 1.04E−10 | −0.0014 (5E−04) | 2.95E−03 | −0.0014 (2E−4) | 1.15E−12 | 37.9 | --+----++ | 0.06 |
| cg21161138 | 5 | 399360 | AHRR | −0.0011 (2E−04) | 1.81E−09 | −0.001 (3E−04) | 9.01E−04 | −0.0011 (2E−4) | 6.66E−12 | 52.3 | ---------- | 0.47 |
| cg15928106 | 7 | 130646078 | FLJ43663 | 0.0014 (2E−04) | 1.14E−06 | 0.0022 (7E−04) | 1.60E−03 | 0.0015 (3E−4) | 1.59E−08 | 52.3 | +-+++-+++++---- | 0 |
| cg11550064 | 2 | 240148191 | HDAC4 | 0.0007 (2E−04) | 1.41E−06 | 0.0006 (6E−04) | 4.06E−03 | 0.0007 (1E−4) | 2.11E−08 | 47.2 | +++++++++++++++ | 0.01 |
| cg09935388 | 1 | 92947588 | GFI1 | −0.0012 (2E−04) | 7.78E−07 | −0.0013 (5E−04) | 8.89E−03 | −0.0012 (2E−4) | 2.32E−08 | 36.9 | ----------++ | 0.06 |
| cg20228731 | 7 | 130646051 | FLJ43663 | 0.0014 (3E−04) | 1.14E−06 | 0.0017 (7E−04) | 9.69E−03 | 0.0015 (3E−4) | 3.87E−08 | 48.4 | ++++++-+++++++ | 0.01 |
| cg06126421 | 6 | 30720080 | NA | −0.0009 (2E−04) | 1.15E−04 | −0.0021 (5E−04) | 7.91E−06 | −0.0011 (2E−4) | 4.50E−08 | 43.8 | --+--+-?- | 0.03 |
| cg14476101 | 1 | 120255992 | PHGDH | 0.0011 (2E−04) | 4.41E−06 | 0.0015 (5E−04) | 2.21E−03 | 0.0011 (2E−4) | 0.0 | | -+++++++++++++++ | 0.99 |
| cg23916896 | 5 | 368804 | AHRR | −0.0011 (2E−04) | 4.34E−05 | −0.0019 (5E−04) | 8.55E−05 | −0.0013 (2E−4) | 4.76E−08 | 0.0 | ---+--+----+ | 0.86 |

The model is adjusted for sex, age, smoking, WBCs, technical covariates, BMI and alcohol consumption. The epigenome-wide significance threshold for association of DNA methylation sites with coffee consumption sets at 1.1×10⁻⁷ (after Bonferroni correction for multiple testing 0.05/450,000).
CpG DNA methylation site, CHR chromosome, Gene annotated gene, NA not annotated, β effect estimate, SE standard error, I2 heterogeneity.

EA participants ($n = 12,868$), none were replicated in AA participants at $P < 0.005$ (0.05/9). Conversely, EWAS in AA participants ($n = 2921$) showed one CpG site (cg05822739) to be associated with coffee consumption ($P = 1.08 \times 10^{-7}$, $\beta = -0.0015$), which was not identified in EA participants despite the larger sample size (Supplementary Table 8).

To minimize the possible confounding effect of smoking on the association between coffee consumption and DNA methylation, we also conducted EWAS on participants who self-reported as never smokers ($n = 2123$). The results of these analyses are shown in Supplementary Table 9. When analysing current and former smokers, the effect sizes did not change substantially compared to the overall sample. In contrast, the results from the subset of never-smokers indicated that the effect sizes of six out of the 11 lead probes, namely: cg09935388, cg21161138, cg25648203, cg06126421, cg05575921, and cg03636183, were increased compared to the beta estimates of the whole meta-analysed samples. As the smoker-status covariate in our study was discrete (never, former or current smoker), the adjusted EWAS model might have been not adequately controlled for smoking exposure (e.g., duration of smoking, amount of smoking for a period of time and second hand smoking). Therefore, we looked up the coffee-associated CpGs in the published EWAS on smoking behaviour (Supplementary Table 10). Eight of the lead coffee-associated CpGs have been previously linked to smoking, suggesting that the residual smoking exposure may remain as a potential confounding factor for the probe associations with coffee consumption.

Among the subjects with low coffee consumption, only one association remained nominally significant (cg14476101, $P = 0.03$). Likewise, we observed that four of the coffee-associated CpGs remained nominally significant in moderate drinkers. When adjusting by tea consumption, the effect estimates of the majority (eight) of the identified CpGs, did not change substantially. We observed a similar pattern (same direction in the effect estimate) for the rest of CpGs that had somewhat change in the effect estimate compared to the main model (Supplementary Table 9). For example, the main model showed a $\beta = -0.001$ for cg21161138 ($P = 6.66 \times 10^{-12}$), while the additional adjustment for tea consumption in the FHS and RS showed a smaller effect ($\beta = -0.0003$). The forest plot for this CpG (Supplementary Fig. 5) showed a negative effect estimate of the CpG in many of the contributing cohorts, which are not in our sensitivity analysis. Hence, we postulate that the observed change of the effect estimates could be the result of this, rather than inadequate adjustment for tea consumption. Additionally, three of the coffee-associated CpGs (cg21161138, cg25648203, and cg03636183) showed nominal significant association with tea consumption (Supplementary Table 11). Lastly, when assessing males and females separately, five CpGs were nominally significant among males and four among females that could be attributed to power, as the effect sizes remained similar compared to the overall sample (Supplementary Table 9).

Of the 11 CpGs significantly associated with coffee consumption, nine have been annotated to the following genes: AHRR, F2RL3, FLJ43663, HDAC4, GFI1, and PHGDH (Fig. 3). A heatmap depicting average expression of these genes across 53 human tissues, available on the "Functional Mapping and Annotation of genetic associations with FUMA" webtool[24], is provided in the Supplementary Fig. 6A, B. Based on the tissue specificity of differential expression using FUMA, PHGDH shows higher relative expression, while HDAC4 and SLC7A11 show moderate expression compare to the other coffee-associated genes in some tissues (Supplementary Fig. 6A). Furthermore, adipose subcutaneous and minor salivary gland show up-regulation of these genes (Supplementary Fig. 7A). The pathway analysis using IPA for

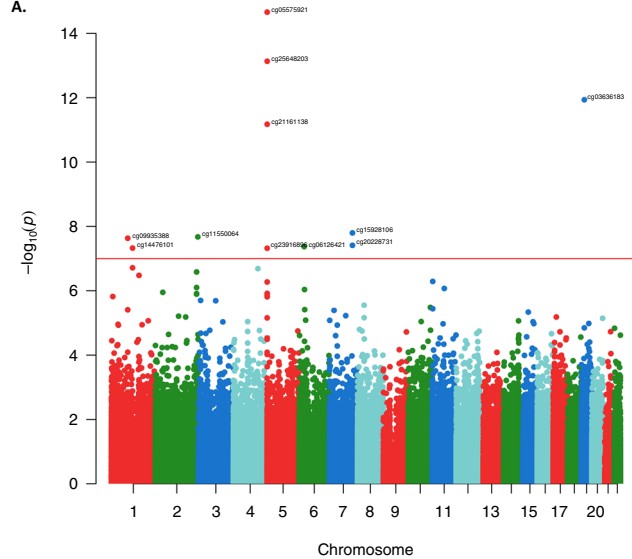

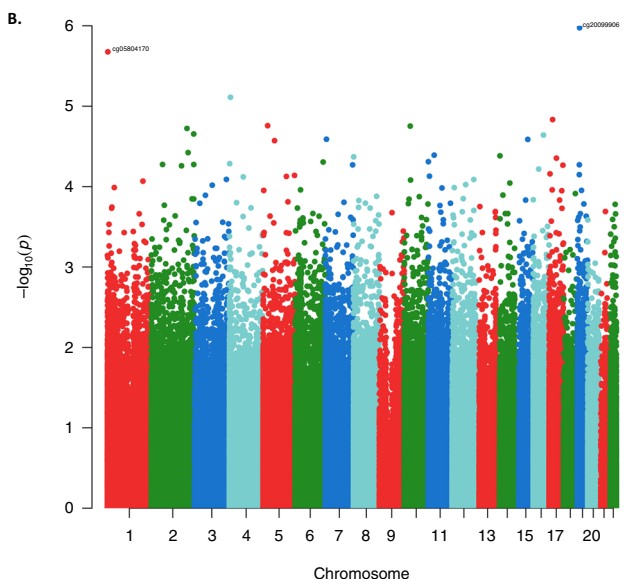

**Fig. 2 Epigenome-wide association study Manhattan plots for coffee and tea consumption.** The plots depict the results of EWAS fixed-effects inverse-variance meta-analysis with the overall sample for coffee (**A**) ($n =$ 15,789) and tea (**B**) consumption ($n = 15,069$) in the fully adjusted model. Each dot corresponds to a single CpG site plotted as the negative logarithm of the $p$-value ($-\log(p$-value) ($y$-axis)) against the genomic position of the CpG site ($x$-axis). The red line indicates the Bonferroni adjusted threshold at epigenome-wide significance $p$-value of $1.1 \times 10^{-7}$ (0.05/450.000).

6 of the annotated genes showed enrichment for Serine biosynthesis ($P = 1.36 \times 10^{-3}$) and Xenobiotic metabolism signalling ($P = 2.71 \times 10^{-3}$) and association with Inflammatory response ($P$-value between $4.48 \times 10^{-2}$ and $4.42 \times 10^{-5}$) (Supplementary Table 12). Pathway analysis for *CACNA1A* and *PRDM16*, the two genes suggestively associated with tea consumption, showed enrichment for white Adipose tissue browning ($P = 3.2 \times 10^{-5}$), nNOS signalling in skeletal muscle cells ($P = 3.6 \times 10^{-3}$) and Maturity onset diabetes of young (MODY) ($P = 5.2 \times 10^{-3}$) (Supplementary Table 13).

The 11 coffee-associated CpGs as well as the two CpGs suggestively associated with tea were further explored in association with genetic variations (meQTL) or expression levels of their nearby or distant genes (eQTM) using the BIOS-BBMRI database. Eight of the coffee CpGs and one of the tea CpGs were

associated with genetic variants in the neighbouring genes (*cis*-meQTLs) (Supplementary Table 14). By overlapping *cis*-meQTL variants with GWAS results in the NHGRI-EBI GWAS Catalogue, we did not find *cis*-meQTLs or their proxies (LD $R^2 > 0.8$) to be associated with any traits in previous GWASs. Furthermore, 6 of the 11 coffee-associated CpGs showed relationship with expression levels of their nearby genes (eQTM) (Supplementary Table 14), including 3 CpGs annotated to *AHRR* that were associated with expression of *EXOC3*. The most significant association with eQTM was between cg14476101 and expression levels of *PHGDH* ($P = 2.05 \times 10^{-55}$). Literature search for the association between the 11–coffee CpGs and any phenotypes or diseases in PubMed showed overlap with some traits that are shown in Supplementary Table 15 and Supplementary Fig. 8. In particular, the CpGs annotated to *AHRR* have been associated with smoking in multiple studies. We did not find eQTM for the two CpGs potentially linked to tea consumption.

We next assessed the causal association between coffee consumption and the 11 identified CpGs in the RS and FHS. The weighted GRS-based MR analysis did not support the causal association, which might be due to lack of statistical power. For example, we observed non-significant results between coffee consumption and cg14476101 (GRS-β $= -3.42 \times 10^{-5}$, GRS-P $=$ 0.22). Additionally, our results from the multi-IVs, conventional and sensitivity MR analyses for this CpG also did not show significant evidence for causality (IVW-β $= 0.01$, IVW-P $= 0.1$) (Supplementary Table 16 and Supplementary Fig. 9).

We also tested the potential causal association of the coffee-associated CpGs with cardiovascular disease and metabolic traits. Multi-instrument MR analyses showed that cg01940273 could be causally associated with T2D, BMI, WHR, LDL-C and total cholesterol (Supplementary Fig. 10); cg05575921 with BMI, WHR and HDL-C (Supplementary Fig. 11); cg09935388 with T2D and HDL-C (Supplementary Fig. 12); cg11550064 with BMI, WHR, HDL-C, LDL-C, total cholesterol, triglycerides and CHD (Supplementary Fig. 13); and cg23916896 with T2D, BMI, HDL-C and total cholesterol (Supplementary Fig. 14 and Supplementary Table 17). The causal association between cg14476101 and fatty liver disease has been previously confirmed by the MR analysis in FHS, where hypermethylation at the locus was associated with lower fatty liver risk ($P = 0.01$)[25].

The inverse association of coffee consumption with liver diseases has been well documented by different researchers[13,26,27]. The CpG cg14476101 and its annotated gene (*PHGDH*) have been reported in previous studies to be associated with fatty liver disease and adiposity[28,29]. Moreover, methylation-gene expression association between cg14476101 and *PHGDH* has been verified in liver tissue[29]. The expression level of the *PHGDH* gene is shown to be associated with liver fat[25]. Also, an intronic genetic polymorphism in *PHGDH* (rs454510, $P = 3 \times 10^{-6}$) has been linked to alcohol-related liver cirrhosis[30]. Thus, we sought to identify a three-way association between coffee consumption, DNA methylation of cg14476101 and liver function in the Rotterdam Study (Supplementary Fig. 2). To this end, we tested the association of coffee consumption and three liver enzymes ($n = 4756$) adjusted for potential confounders, which showed a negative association between coffee consumption and serum levels of AST ($P = 0.008$, β $= -0.005$) and GGT ($P = 0.004$, β $= -0.011$) (Supplementary Table 18). In addition, we tested the association of DNA methylation at cg14476101 with the liver enzymes ($n = 1406$) adjusted for age, sex, BMI, smoking, whole blood cells proportion, batch effects and excessive alcohol consumption, and observed a nominal association with the serum levels of AST ($P = 0.016$, β $= -0.26$) and a suggestive association with GGT ($P = 0.06$, β $= -0.43$). These data suggest that the link between coffee consumption and fatty liver disease could be mediated by

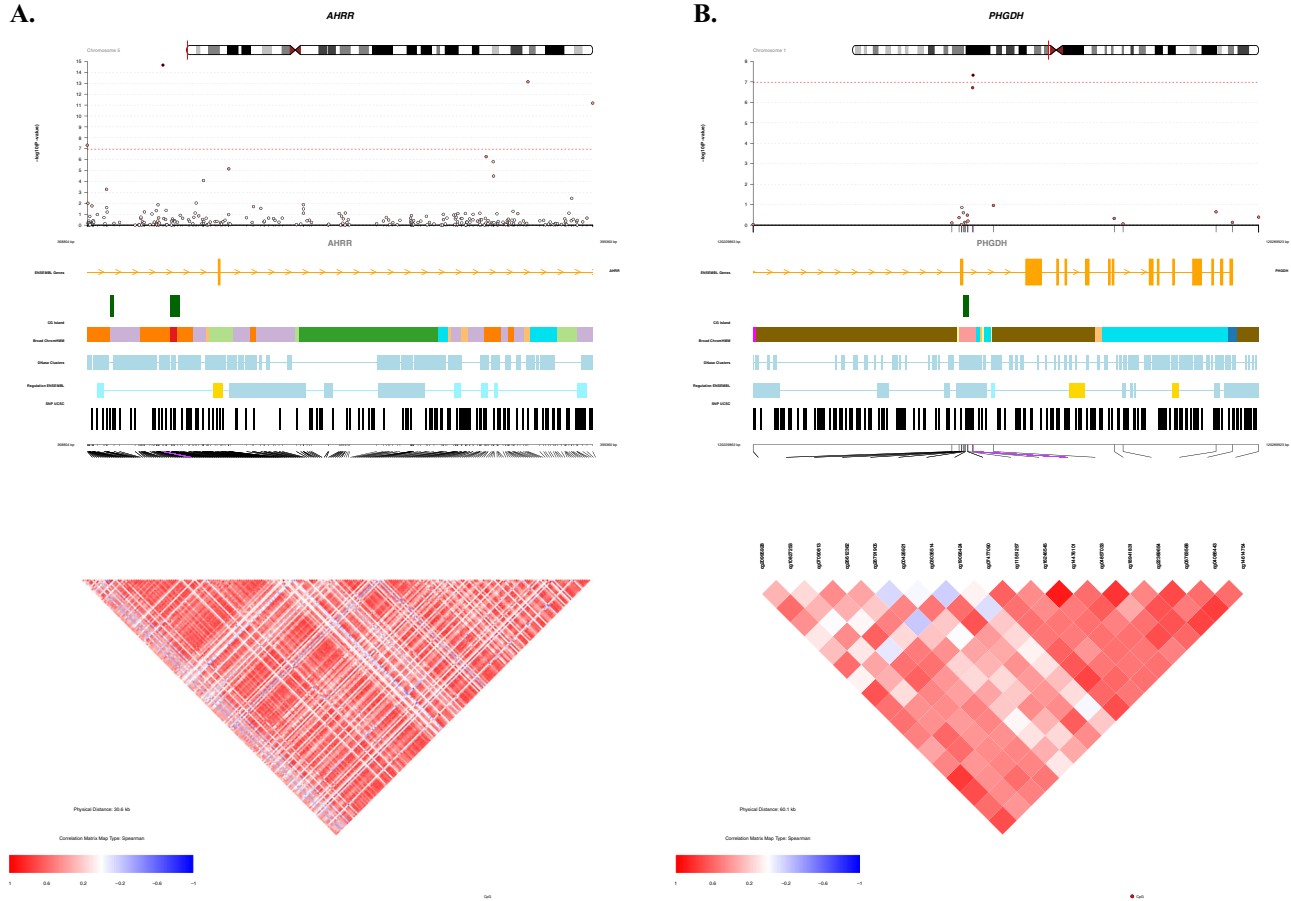

**Fig. 3 The CoMET plots depicting genomic regions where the CpGs annotated to *AHRR* (A) and *PHGDH* (B) are located.** The *x*-axis indicates the position in base pair (bp) (hg19) for the region, while *y*-axis indicates the strength of association from EWAS with coffee consumption. The red line indicates the Bonferroni threshold for epigenome-wide significance ($P = 1.1 \times 10^{-7}$). The figure was computed using the R-based package CoMET, while the Ensembl is a genome database resource (http://ensemblgenomes.org/). The correlation of the surrounding CpGs was computed using methylation measures in the Rotterdam Study.

**Table 3 Inverse-variance weighted fixed effects meta-analysis of EWAS with tea consumption.**

| CpG | CHR | Position | Gene | Overall meta-analysis ($n = 15,069$) | | | | |
|-----|-----|----------|------|-------------|---------|-------|-----------|--------|
| | | | | *B* (SE) | *P*-value | $I^2$ | Direction | Het P-v |
| cg20099906 | 19 | 13344820 | *CACNA1A* | −0.0008 (2E−04) | 1.06E−06 | 17.7 | -+-+--++--+--+-- | 0.25 |
| cg05804170 | 1 | 3121514 | *PRDM16* | −0.0002 (2E−04) | 2.11E−06 | 0 | -+--+-+----++- | 0.98 |

The model is adjusted for sex, age, smoking, WBCs, technical covariates, BMI, and alcohol consumption. The epigenome-wide significance threshold for association of DNA methylation sites with tea consumption sets at $1.1 \times 10^{-7}$. The table depicts two CpGs suggestively associated with tea consumption with a borderline *p*-value between $2.0 \times 10^{-6}$ and $1.1 \times 10^{-7}$.
*CpG* DNA methylation site, *CHR* chromosome, *Gene* annotated gene, *β* effect estimate, *I2* heterogeneity.

*PHGDH* expression via altering DNA methylation levels at cg14476101.

To gain further insight into the biological mechanism linking *PHGDH* to fatty liver disease, we conducted experimental studies. We measured the expression level of *PHGDH* in several human liver cell lines and, subsequently, related it to the expression levels of a panel of lipid-associated genes. Figure 4a displays the *PHGDH* expression level in seven liver cell lines. From this, we selected SNU398 cells, with the highest expression levels of *PHGDH*, and SNU449 cells, with the lowest expression levels of *PHGDH*, and compared the relative expression levels of *PHGDH* with nine known lipid-related genes, reported in the previous GWAS and experimental studies to be involved in lipid metabolism[31–33]. The *PHGDH* expression level was correlated

with the expression levels of five of these lipid-associated genes (Fig. 4b). Next, we knocked down the *PHGDH* expression in PLC/PRF/5 cells using lentiviral shRNA vectors (Fig. 4c). After silencing *PHGDH*, we observed a significant decrease of the *LPL* expression and a significant increase of the *LDR* and *ABCA1* expression ($P < 0.05$) in both knocked down cells (Fig. 4d), in line with the observed correlations in SNU398 and SNU449 cells. These results suggest a potential role of *PHGDH* in lipid metabolism and fat accumulation in the liver, which could occur through regulating the expression of lipid-associated genes.

## Discussion

In this study, we conducted the largest EWAS meta-analyses of coffee and tea consumption to date comprising more than 15,000

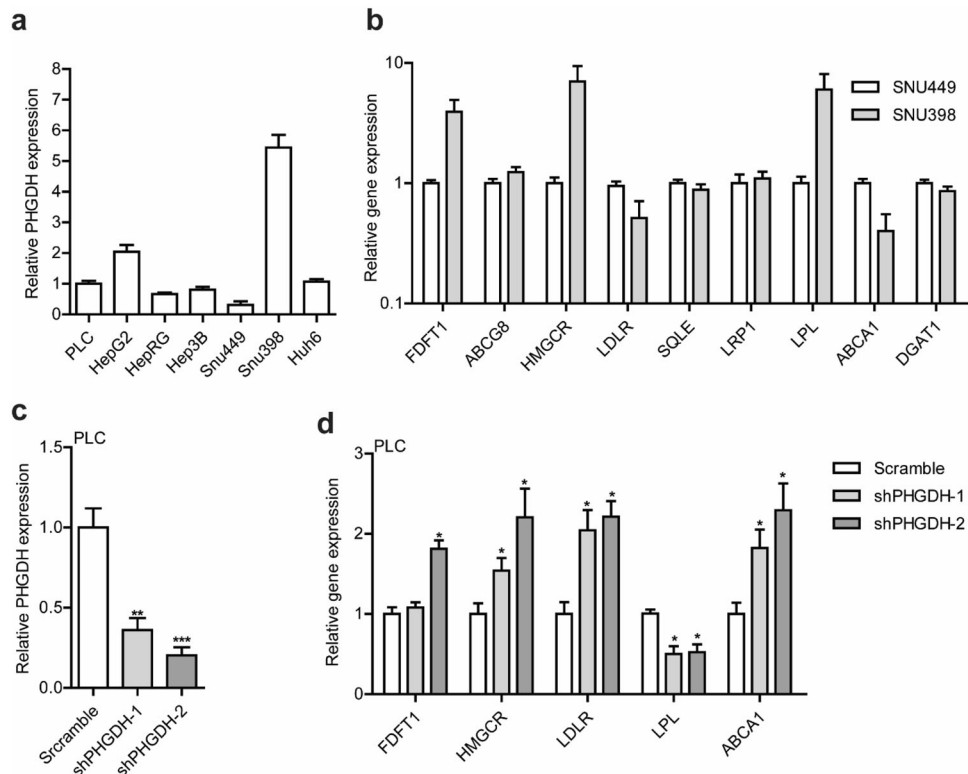

**Fig. 4 *PHGDH* gene expression levels in liver cell lines and relative to expression levels of lipid-associated genes. a** Relative expression levels of *PHGDH* against a reference gene (*GAPDH*) in 7 human liver cell lines. Gene expression levels were quantified by qRT-PCR. Data were normalized to the PLC cell line (PLC, set as 1). **b** Relative expression levels of 9 lipid-associated genes in SNU499 cell line (with the lowest level of *PHGDH* expression) and SNU398 cell line (with the highest level of *PHGDH* expression) are shown. Relative gene expression levels were quantified by qRT-PCR. *GAPDH* serves as a reference gene, and gene expression levels in SNU449 cell line set as 1. This figure shows that, compared with SNU449 cells, SNU398 cells differentially express five of the lipid-associated genes (*FDFT1, HMGCR, LDLR, LPL,* and *ABCA1*). **c** Established *PHGHD* knockdown cell lines (shPHGHD-1 and -2), PLC cells transduced with lentiviral shRNA vectors targeting *PHGDH* or scramble control. qRT-PCR analysis of *PHGDH* expression were performed in stable knockdown or scramble control PLC cells. Data are normalized to the scramble control (scramble, set as 1). **d** Expression levels of five lipid-associated genes in stable *PHGDH* knockdown or scramble control PLC cells. Data were normalized to the scramble control (scramble, set as 1). The figure demonstrates that knockdown of *PHGDH* gene expression by lentiviral shRNA vectors resulted in significant decrease in the expression level of *LPL* and significant increase in the expression levels of *LDLR* and *ABCA1* in both knockdown cells. Data in the figures are presented as mean values ± SEM of $n = 3$ biologically independent experiments. The Mann–Whitney *U*-test (two-sided) was used to compare differences between two independent groups. Differences were considered significant at $P < 0.05$, which indicated by * (**$P < 0.01$ and ***$P < 0.001$).

participants. We found that coffee consumption was associated with differential methylation of peripheral blood-derived DNA at 11 CpG sites, while tea consumption was not significantly associated with changes in DNA methylation. The genes annotated to some of the coffee-associated CpGs have potential relevance in pathways underlying coffee metabolism and have been linked to different health outcomes[34–36].

Seven of the 11 coffee CpGs were inversely associated (i.e. higher coffee consumption was associated with lower levels of DNA methylation in these CpGs), four of which are annotated to the Aryl-Hydrocarbon Receptor Repressor (*AHRR*). This gene encodes a repressor of *AHR*, which itself has been previously associated with coffee consumption in a large-scale GWAS ($n = 91,462$)[37]. Both *AHRR* and *AHR* genes, alongside with *CYP1A1* and *CYP1A2*, belong to the Xenobiotic metabolism pathway, in which *AHR* is activated via various ligands such as polycyclic aromatic hydrocarbons (PAHs) or with ligands of natural origins (e.g., food)[38]. PAHs are carcinogenic substances found in tobacco smoke[39], which can also be formed during coffee roasting processes[40]. It has been established that PAHs mediate their impact on the cell via *AHR*[41]. Nevertheless, the *AHRR* CpGs identified in our analysis associated with coffee consumption have been reported in previous EWAS of smoking[42,43]. When we

restricted our analysis to never smokers, the effect sizes of six out of the 11 lead probes were increased. The differences in the effect sizes and the overlap of some of these coffee-CpGs with findings from previous large-scale EWAS on smoking may indicate that our results could be yet affected by some residual confounding of smoking. Although we accounted for smoking status in our analyses, given that cigarette smoking is associated with coffee consumption[44] and smoking has a notable effect on DNA methylation[42], the association between these CpGs with coffee consumption might warrant cautious interpretation. We speculate that methylation at the *AHRR* locus might influence the Xenobiotic pathway via functionally related genes (*AHR, CYP1A1* and *CYP2A2*) identified previously in the coffee consumption GWAS[37]. We also found that the four CpGs annotated to *AHRR* are associated with expression of a neighbouring gene, *EXOC3* in blood. However, due to tissue-specific expression patterns of genes, it is still possible to see eQTM for *AHRR* gene in a different tissue (e.g. liver) that needs to be investigated in future studies.

The other three CpGs inversely associated with coffee consumption in our study are annotated to *F2RL3, GFI1* and *IER3*. These genes are involved in a wide range of phenotypes from inflammatory response[45,46], to cancer[47,48] and cardio-metabolic diseases[49,50]. Because of the strong association between coffee

intake and cardiovascular disease[51] as well as cancer[52], further experimental studies are warranted to show whether alteration of DNA methylation levels at the identified CpGs could change gene expression and confer risk for these complex diseases.

Besides, five of the 11 CpGs were positively associated with coffee consumption. Four of these are annotated to *FLJ43663*, *HDAC4* and *PHGDH*. To date, little research has shown the link between *FLJ43663* and disease predisposition. A recent study showed the involvement of *FLJ43663* as a risk factor for Behçet's disease[53], while another study reported *FLJ43663* gene polymorphisms to be associated with risk of breast cancer in a Han Chinese population[54]. The second gene, *HDAC4* encodes histone deacetylase with a molecular role of deacetylation of lysine residues of the core histones[55]. As histone modifications are another known epigenetic mechanism, this might be an indication of the potential interplay between two epigenetic modifications. Moreover, *HDAC4* gene is involved in cocaine-related behaviours[56], with a proposed mechanism that cocaine-induced nuclear export of *HDAC4* could promote development of cocaine reward behaviours. Furthermore, animal studies have shown that *HDAC4* inhibition increases sensitivity to cocaine, whereas overexpression of the gene has opposite effect, which further supports the importance of *HDAC4* in conditioned place preference in addictive behaviour[57]. It is interesting to note that *HDAC4* and *HDAC5* belong to the same *HDAC* classification[55], and *HDAC5* gene has been associated previously with cocaine dependence in recent EWAS of cocaine and crack dependents[58].

Among all genes annotated to the 11 coffee-associated CpGs, *PHGDH* is of particular interest. This gene encodes the phosphoglycerate dehydrogenase enzyme that catalyses the first and rate-limiting step in the phosphorylated pathway of serine biosynthesis. A methylation-gene expression association between cg14476101 and *PHGDH* has been demonstrated in blood and liver[29]. The CpG site has been reported to be negatively associated with the levels of liver enzymes in serum[28] and the risk of non-alcoholic fatty liver disease (NAFLD)[25]. Furthermore, methylation of cg14476101 has been linked to adiposity measured by BMI and waist circumference[29]. Another EWAS has also reported the association between this CpG and blood concentration of steroid hormones, which are upregulated in obesity[59]. In line with our findings in the Rotterdam Study, several studies have shown an inverse association between coffee consumption and liver enzymes, including ALT, AST, and GGT[60]. Moreover, previous studies have associated coffee consumption with reduced risk of chronic liver disease[26], hepatocellular carcinoma[61], and cirrhosis[62]. Together, these findings strongly proposed a link between the expression of *PHGDH* gene and liver function that can be modified by coffee intake via altering the DNA methylation levels of cg14476101.

We attempted to provide additional evidence of directionality of the effect of coffee consumption on cg14476101 using MR methods; however, the power of our MR analyses was attenuated due to the small sample size of our DNA methylation data. The lack of strong genetic instruments for coffee exposure might also be an important determinant of the results observed. In line with this reasoning, a recent review of the MR studies on coffee and caffeine consumption[63] has indicated the need of substantial sample sizes to assess the causality when it comes to coffee. The non-significant results from our MR analyses could also be explained by some other reasons. Firstly, population stratification, albeit we used genetic information from a large GWAS performed mainly in European population and adjusted by population substructure[64]. Secondly, pleiotropy, since some of the SNPs used as IVs have also been associated with lipid traits and body size which might influence the causal estimates. Yet the results of MR excluding potential pleiotropic variants were fairly similar and

MR-Egger, implemented in this study, is an useful approach to account for pleiotropy[65]. Thirdly, the genetic variants for coffee consumption were associated with number of cups of coffee per day among coffee drinkers, and the effect estimates might not relate to DNA methylation observed among ever/never coffee drinkers[66]. An alternative conclusion could be that the methylation patterns reflect pathways of coffee consumption effects on the body, but are not necessarily causal. As the analyses have been explored using meQTL data obtained from blood, we cannot rule out that the potential differences might be observed in other relevant tissues (e.g., liver). Unfortunately, important resources have not reported meQTL data from other tissues different than blood and this may warrant further investigation in future studies.

We additionally knocked down the expression of *PHGDH* in human liver cells and revealed a correlation between expression of this gene and some lipid-associated genes (*LPL*, *LDLR* and *ABCA1*), suggesting a potential role of *PHGDH* in hepatic-lipid metabolism. In this line, previous GWAS have associated SNPs near *PHGDH* with serum levels of total cholesterol[67], some metabolites[68] and metabolic traits[30,69]. Also, previous evidence has indicated that a reduced expression of *PHGDH* is linked to the development of fatty liver disease[36]. In an independent study, we previously showed that a CpG (cg06690548) annotated to *SLC7A11* is associated with liver enzymes and NAFLD[25,28]. Then, performing similar knock-down experiments we demonstrated the involvement of *SLC7A11* in hepatic-lipid metabolism[28]. The *SLC7A11* gene is known as transporter of cysteine and glutamate, whereas caffeine promotes glutamate release in the posterior hypothalamus[70]. The same CpG (cg06690548) showed a borderline association ($\beta = 0.0008$, $P = 2.0 \times 10^{-7}$) with coffee consumption here. Thus, more experimental studies are merited to further elucidate in what way epigenetic modification of *PHGDH* and *SLC7A11* could explain the beneficial effect of coffee consumption on lipid metabolism and liver diseases.

Our EWAS meta-analysis of tea consumption showed no significant association, despite having the advantage of much larger sample size compared to the previously published EWAS[21]. When including coffee consumption in the adjusted EWAS model of tea, in the earlier study, Ek *et al.* reported two CpGs associated with tea consumption in women, nevertheless, those CpGs were not replicated in our study[21]. The lack of statistically significant associations between tea consumption compared with coffee in our analysis can be explained by specific nutrients that are present only in coffee (or present in higher abundance compared to tea) such as caffeine or various phenolic compounds[5]. Moreover, data collection of tea consumption might introduce more heterogeneity compared to coffee. It is also worth mentioning that there is extensive literature on GWAS with coffee consumption (Supplementary Table 5), but only one with tea consumption (in Japanese population)[71]. Although we cannot deduce with certainty the reasoning behind this, one might suspect the weaker effect of tea consumption and potential existence of publication bias due to the lack of non-significant findings. In the future, perhaps bigger sample size would provide more understanding in tea consumption EWAS. Given the well-powered EWAS conducted here, it is also possible that tea consumption has no discernible impact on peripheral blood DNA methylation.

DNA methylation can differ across continental ancestries[72], challenging replication across populations of varying descent in epigenetic studies[73,74]. Here we observed this discrepancy in our results for some CpGs. In particular, none of the coffee-associated CpGs in EA cohorts were replicated in AA. Moreover, the AA participants showed one CpG site (cg05822739) associated with coffee consumption that was not replicated in EA. These

discrepancies between continental ancestries may be explained by the following reasons. First, some CpGs might be ancestry-specific. Second, given the strong influence of culture on coffee drinking habit, its frequency may differ among different ancestry groups. For example, in the EA cohorts, the mean coffee consumption intake was 2.3 cups/day, while in the AA cohorts this number dropped to 0.79 cups/day. Finally, the difference in sample sizes in our meta-analysis (EA = 13,146; AA = 2921) might also play a role in the discrepancies observed among the populations.

The main strengths of this study are the large sample size and multi-ethnic contribution. All contributing cohorts had DNA methylation measured in whole blood, and adjustment for white blood cell heterogeneity allowed us to account for different epigenetic markers within cells present in the blood. Incorporation of different adjustment models also allowed us to limit confounding to a certain extent. The findings of this study should also be considered in light of some limitations. One important concern regarding this analysis is smoking, given that the effect of smoking on DNA methylation has been recognized[42] and previous studies have shown that heavier smokers tend to drink more coffee[44]. Even though we adjusted for smoking in our analysis, there is likely to be some residual confounding. Furthermore, smoking status might be misclassified or the possibility of second-hand smoke cannot be ruled out. The results presented could reflect potential pleiotropy, confounding or both, or it could provide insight into the potential causal role of coffee on DNA methylation and disentangling these would merit further investigations. In addition, coffee was assessed and used as a continuous variable (cups/day), and some cohorts have different cup sizes. However, we believe that this limitation would rather dilute the findings, attenuating any association rather than falsely inflating it. We also did not include information on coffee brewing methods, which might have a large effect on what compounds are in the final beverage (e.g. filtered vs non-filtered coffee). It is also important to address the potential limitation of this study resulting from the method of dietary data collection via FFQ. While the FFQ are the most cost-effective way to collect the dietary data, they are subject to some inaccuracies (e.g. participants might have a recall bias when filling in the questionnaires). Nevertheless, when compared to the other dietary assessment methods (e.g. 24-h recall or food diary), the FFQ usually reports a lower within-person variation[75]. Finally, since our study consists mainly of middle aged and elderly individuals of two ancestries, other studies are needed to assess the generalizability of our findings to other age groups and ancestries.

In summary, we found that coffee consumption is significantly associated with differential DNA methylation at multiple CpG sites, the genes annotated to some of these CpGs are involved in pathways underlying coffee metabolism. Our findings may provide insights into the mechanism of action of coffee intake in conferring risk of diseases. Future studies are warranted to further explore the biological relevance of the associated DNA methylation sites and genes in relation to different health outcomes.

## Methods

**Study population.** This study was conducted within the framework of the Cohort for Heart and Aging Research in Genomic Epidemiology (CHARGE consortium)[76] and additional participating cohorts, resulting in a total sample size of 15,789 participants. Clinical characteristics of the participants included in our study are presented in Table 1. The discovery phase included 9612 participants of European ancestry (EA) from the following cohorts (listed in alphabetical order): Airwave[77], Avon Longitudinal Study of Parents and Children (ALSPAC)[78], two independent datasets from the ESTHER Study (ESTHER_a and ESTHER_b)[79], Framingham Heart Study (FHS)[80], Cooperative Health Research in the Augsburg Region Study (KORA)[81], two cohorts of the Rotterdam Study (RS-II and RS-III)[82] and TwinsUK[83]. We sought replication of the associated CpG sites from the discovery phase, in an independent population consisting of 6,177 participants of European

(EA) and African American (AA) ancestries (18.3%). The replication phase included two ethnically different sub cohorts of Atherosclerosis Risk in Communities Study (ARIC_EA and ARIC_AA)[84], two ethnically different sub cohorts from the Cardiovascular Health Study (CHS_EA and CHS_AA)[85], and two independent studies from the European Prospective Investigation into Cancer and Nutrition (EPIC_Italy and EPIC_IARC)[86]. All participants provided written informed consent, and all contributing cohorts confirmed compliance with their local research ethics committees or Institutional Review Boards. Detailed information of the participating cohorts are provided in Supplementary Information.

**Assessment of coffee and tea consumption.** Data on coffee and tea intake was collected either by interview or using food frequency questionnaires (FFQs). As some FFQs collected beverage intake over different periods of time (monthly, weekly or daily), data was harmonized among the cohorts to cups per day by taking the average intake of coffee/tea over the period of time specified by the FFQ utilized by each cohort. For instance, if the beverage consumption was collected over the period of one month, daily consumption was estimated from the available data and multiplied with the frequency of consumption. Furthermore, if the beverage intake data was collected categorically, the median value was taken from the available data (e.g. 2.5 cups/day was used for the 2-3 cups/day category). If applicable, we excluded herbal tea and others, as green and black tea are derived from the different processing and harvesting of leaves from the same plant -*Camellia sinensis*[10]. Herbal tea does not contain any caffeine and green tea contains approximately half the caffeine compared to black tea (3.1 mg/fluid ounce, 5.9 mg/fluid ounce)[37]. In a subset of cohorts (RS-III-2, ALSPAC, EPIC_IARC, CHS_EA and CHS_AA), coffee and tea consumption data were collected a few years prior to the collection of whole blood, from which DNA methylation data was measured. Due to evidence from a previous research showing that coffee and tea consumption tend to be stable over longer periods of time[87], we used these data for our analysis.

**DNA methylation profiling.** All participating cohorts measured DNA methylation in peripheral blood using the Infinium Human Methylation 450K Bead-Chip (Ilumina, San Diego, CA, USA) except Airwave cohort, where the Infinium Methylation EPIC (850K) Bead-Chip was used[88]. DNA methylation status was calculated with the $\beta$-value, signal from the methylated probe divided by the overall signal intensity. The methylation percentage of CpG sites was reported as a continuous $\beta$-value range between 0 (no methylation) and 1 (full methylation). Additional details are outlined in Supplementary Information. Cohort specific methods of normalization are shown in Supplementary Table 1.

**EWAS of coffee and tea consumption.** DNA methylation was considered as the dependent variable with coffee or tea consumption each as predictors of interest. Conventionally, each participating cohort performed an EWAS as a set of mixed effects linear-regression models, one CpG site at a time. In total, two linear mixed effects regression models were computed for each of the two exposures of interest. In the basic model (Model 1): we included age, sex, smoking status (never, former, and current), white blood cells (either measured or imputed based on the Houseman algorithm[89]) as fixed effects, and technical covariates as random effects to control for batch effects. In the second model (Model 2), we additionally adjusted for body mass index (BMI, kg/m$^2$) and alcohol consumption (g/day). The findings from Model 2 were considered as the primary results. All potential confounders were collected at the same time point of blood sampling for DNA methylation. Genetic principal components were included as covariates to account for population stratification, if required. A detailed description of the covariates included in the models by each cohort is provided in Supplementary Information. Tea and coffee consumption were added as covariates to Model 2 of EWAS on coffee and tea, respectively, in order to assess potential confounding effects on DNA methylation. This analysis was conducted on FHS, the largest cohort, and the RS.

**EWAS meta-analysis.** Since data originated from multiple sources, we performed quality control (QC) centrally. Each participating cohort submitted the EWAS summary statistics for the QC followed by meta-analysis. For this step, we used a specific package in R developed for QC within EWAS, namely "QCEWAS"[90]. We computed the genomic inflation factor (lambda) and checked quantile-quantile (QQ) plots for Model 2 for both coffee and tea consumption EWASs. Additionally, we computed effect-size distribution plots to assess the effect-size scale of each participating cohort. Prior to their inclusion into the meta-analysis, all probes with a SNP, non-CpG probes and cross-reactive probes were removed as suggested by Chen et al. [91]. Results across independent cohorts were combined in both discovery and replication phase by using inverse variance fixed effects meta-analysis, implemented in METAL v.2011-03-25[92]. Moreover, we assessed heterogeneity of effect estimates among cohorts using Cochran's $Q$-test for heterogeneity implemented in METAL[92]. If there was nominal evidence for heterogeneity ($P < 0.05$), we performed random-effects inverse-variance meta-analysis using the method implemented in GWAMA[93]. We created Manhattan plots using the qqman package in R. In addition, we stratified two of the largest cohorts (FHS and RS) by sex, smoking status and coffee consumption frequency. Smoking status was evaluated on current vs never smokers; coffee consumption frequency was determined

as none or infrequent drinkers (<1 cup/day), moderate drinkers (>=1 and <4 cup/day) and high or frequent drinkers (>=4 cup/day) using a large-scale GWAS on coffee consumption as a reference for categorization of the variable[37].

The discovery EWAS meta-analysis was conducted in 9,612 EA participants and differentially methylated CpGs at the suggestive threshold of $P < 5.0 \times 10^{-6}$ were interrogated. The CpGs that passed this threshold were tested for replication in the independent panel comprising 6177 participants of European and African American ancestries, using the same models as implemented in the discovery phase and with a Bonferroni corrected p-value threshold, defined as 0.05 divided by the number of associated CpGs in the discovery phase. The significantly associated CpGs were retrieved from the combined meta-analysis with the whole samples at the epigenome-wide significance threshold ($P < 1.1 \times 10^{-7}$). If the CpG was missing from more than 4 participating cohorts, it was removed from the analysis. Forest plots of the study-specific effect estimates were computed for significantly associated CpG sites using the metaviz package (https://github.com/Mkossmeier/metaviz) in R environment.

Due to potential discrepancy in DNA methylation patterns between different ethnicities, we conducted meta-analysis EWASs separately in EA ($n = 12,868$) and AA ($n = 2921$) participants. We first examined whether the significantly associated methylation sites in EA participants passed the Bonferroni-corrected p-value threshold in the AA participants. Next, we tested whether the significant methylation sites in the AA participants replicated in the meta-analysis of the EA participants.

Furthermore, we examined the potential impact of time varying exposure in cohorts that had different time points for methylation and coffee/tea consumption data collection. For this analysis, we excluded four cohorts (RS-III-2, ALSPAC, CHS_EA and CHS_AA) with different time points of data collection from the overall sample and meta-analysed the remaining cohorts.

**Integration of EWAS results with genetic variation and gene expression.** DNA methylation may have an impact on the transcription of genes; hence we used genetic variants and gene expression data from five Dutch biobanks (BIOS-BBMRI database) in a total of 3841 whole blood samples (http://www.genenetwork.nl/biosqtlbrowser/), and explored whether DNA methylation levels of the significant CpGs affect expression levels of their annotated/nearby genes (cis-expression quantitative trait methylation (eQTM). The BIOS-BBMRI database was also used to seek genetic variants influencing methylation levels of nearby or far-away genes (cis- and trans-methylation quantitative trait loci (meQTL).

**Functional and regulatory annotation of CpG sites.** We conducted hypergeometric tests with a Bonferroni correction to compare the genomic characteristics of the replicated CpGs with the whole set of analysed CpGs, using the Infinium Human Methylation 450 Bead-Chip annotation files. Further, we queried cis-meQTLs in the platform of Functional Mapping and Annotation of Genome-Wide Association Studies (FUMA GWAS)[24]. Using this platform, we examined the overlap between cis-meQTLs with signals in the NHGRI-EBI Catalogue of published GWAS[94]. As epigenetic signatures are tissue dependent, and our analysis was limited in blood samples, we used the GTEx expression database - which provides an insight into differential expression of relevant genes across different human tissues. For this analysis, we used genes annotated by Illumina 450 K (or the nearest gene) to the significantly associated CpGs. Moreover, we searched PubMed using the 'CpG id' as the keyword to search any potential links between significant CpGs and a range of health outcomes. Pathway analysis for the annotated genes was also performed using the IPA software (https://www.qiagenbioinformatics.com/products/ingenuity-pathway-analysis/).

**Mendelian randomization (MR) study.** We implemented a two-sample MR approach to evaluate the potential causal effect of coffee consumption on the identified CpGs, investigating whether the DNA methylation changes are a consequence of coffee consumption (Supplementary Fig. 1). To this end, we used 50 independent SNPs, reported in previous GWASs on coffee consumption, as instrumental variables (IVs) (Supplementary Table 2)[64,95].

In addition, we assessed the potential causal association of coffee-related CpGs with a number of cardiovascular and metabolic traits, including coronary heart disease (CHD), T2D, BMI, waist–hip ratio (WHR), lipid traits (HDL-C, LDL-C, total cholesterol, triglycerides), and fatty liver disease. For each CpG, we calculated IVs for DNA methylation levels based on methylation quantitative trait loci (cis-meQTL) obtained from FHS cohort ($N \sim 4170$)[96]. Two methods were used to explore causality. First, a weighted genetic risk score (GRS) was constructed for coffee consumption. The other MR approaches implemented were the inverse variance weighting (IVW) method, and sensitivity MR analyses: the weighted median and MR-Egger methods. We used MR-PRESSO (MR pleiotropy residual sum and outlier) to identify horizontal pleiotropic outliers in multi-instrument summary-level MR testing (https://github.com/rondolab/MR-PRESSO)[97]. All MR methods for multiple genetic instruments were conducted using the statistical "MendelianRandomization" R-package[98]. Additional information of the MR methods implemented in this study is outlined in Supplementary Information.

**Association of coffee consumption, DNA methylation and liver function.** Due to the well-documented association between coffee consumption and liver

function[13,26], we also ran a three-way association to assess the correlation of a coffee-associated CpG with liver enzymes and fatty liver disease in the Rotterdam Study (Supplementary Fig. 2). We first tested the cross-sectional associations between coffee consumption and liver enzymes in the Rotterdam Study ($n = 5192$). Serum GGT, ALT and AST levels were determined using Merck Diagnostica kit on an Elan Autoanalyzer (Merck, Darmstadt, Germany). The liver enzymes were log transformed to obtain normal distribution. Linear regression models were implemented where each liver enzyme was an outcome, and the main exposure was coffee consumption (cups/day) adjusted for sex, age, smoking, BMI and excessive alcohol consumption. Excessive alcohol consumption was defined as >14 units/week for women and >21 units/week for men. Next, we tested the association of the coffee-related CpG with liver enzymes in the Rotterdam Study ($n = 1406$)[13]. Generalized linear mixed effects models were fitted using the R package lme4 and liver enzymes were log transformed to obtain normal distribution. Three models were analysed, where each liver enzyme was an outcome, adjusted for age, sex, BMI, smoking, whole blood cells proportion, batch effects and excessive alcohol consumption. All analyses were performed using the statistical package R, version 3.0.2.

**Quantitative RT-PCR and knockdown of *PHGDH* in liver cell lines.** Seven established human hepatoma cell lines (including PLC/PRF/5, HepG2, HepRG, Hep3B, SNU398, SNU449 and Huh6) were cultured separately. HepG2, Hep3B, SNU398, SNU449 and Huh6 were cultured in Dulbecco's modified Eagle's medium (Invitrogen-Gibco, Breda, the Netherlands) complemented with 10% (v/v) foetal calf serum (Hyclone, Lonan, UT), 100 IU/ml penicillin, 100 μg/ml streptomycin, and 2 mM L-glutamine (Invitrogen-Gibco). The hepatoblastoma cell line PLC/PRF/5 was cultured on fibronectin/collagen/albumin-coated plates (AthenaES) in Williams E medium (Invitrogen-Gibco, Breda, the Netherlands) complemented with 10% (v/v) foetal calf serum, 100 IU/ml penicillin, 100 μg/ml streptomycin, and 2 mM L-glutamine. The human liver progenitor cell line—HepaRG was cultured in William's E medium supplemented with 10% (v/v) foetal calf serum, 100 IU/ml penicillin, 100 μg/ml streptomycin, 5 μg/ml insulin (Sigma-Aldrich, St. Louis, MO), and 50 μM hydrocortisone hemisuccinate (Sigma-Aldrich, St. Louis, MO). The identity of all cell lines was confirmed by STR genotyping.

RNA was isolated using the Machery-NucleoSpin RNA II kit (Bioke, Leiden, The Netherlands) and quantified using a Nanodrop ND-1000 (Wilmington, DE, USA). cDNA was synthesized from total RNA using a cDNA Synthesis Kit (TAKARA BIO INC). The cDNA of all target genes was amplified for 50 cycles and quantified with a SYBRGreen-based real-time PCR (Applied Biosystems) according to the manufacturer's instructions. *GAPDH* was considered as a reference gene to normalize gene expression. Relative gene expression was normalized to *GAPDH* using the formula $2^{-\Delta\Delta CT}$ ($\Delta\Delta CT = \Delta CTsample - \Delta CTcontrol$). All primer sequences are included in Supplementary Table 3.

Lentiviral pLKO knockdown vectors (Sigma-Aldrich) targeting *PHGDH* or control were obtained from the Erasmus Biomics Center and produced in HEK293T cells. After a pilot study, the shRNA vectors (Supplementary Table 3) exerting optimal gene knockdown were selected. Stable gene knockdown cells were generated after lentiviral vector transduction and puromycin (2.5 μg/ml; Sigma) selection. The relative expression levels of *PHGDH* with nine lipid-associated genes, reported in the previous GWAS and experimental studies[31–33], were examined.

**Reporting summary.** Further information on research design is available in the Nature Research Reporting Summary linked to this article.

## Data availability

The summary statistics from EWAS meta-analysis of coffee and tea consumption will be made available upon publication on the CHARGE dbGaP site under the accession number phs000930. The dataset used to extract genetic variants and gene expression data is based on the BIOS-BBMRI database (freely available at: http://www.genenetwork.nl/biosqtlbrowser/). Functional mapping and annotation was performed using FUMA GWAS (freely available at https://fuma.ctglab.nl/), which also includes GTEx data. All other relevant data supporting the key findings of this study are available within the article and its Supplementary Information files or from the corresponding author upon reasonable request. All the software and programmes used to conduct these analyses are freely available through the links mentioned in the manuscript.

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

## Acknowledgements

The authors are grateful to the staff and participants of all cohorts involved in this study for their important contributions. Specific funding and acknowledgements statements of each study can be found in Supplementary Information.

## Author contributions

M.G., I.K., and M.K. contributed to study design. I.K., E.P., J.M., S.M., D.S., B.K., Y.Z., S.A., G.F., J.H., J.E.C., N.K., B.S., and S.A.G. contributed to cohort-specific data analyses. I.K. contributed to meta-analyses of EWAS. E.P. and J.M. contributed to mQTL and Mendelian Randomization analyses. M.G. and Q.P. designed and Y.L. performed the experimental studies in liver cell lines. T.V., K.B., M.A.I., J.L.T., E.A.H., C.C., P.C.T., R.C., J.M., A.G.U., M.W., D.L., M.K., M.F., M.W., H.B., Z.H., C.R., P.V., P.E., J.T.B., N.S., A.P., A.D., L.F., S.P., N.V., V.C., S.G., and R.J.K. contributed to cohort design and management, and data collection. M.G., IK., and E.P. contributed to interpretation of the results and writing of manuscript. M.G., I.K., E.P., S.M., M.K., J.H., C.M.R., D.L., B.S., H.B., J.L.T., K.W., M.W., S.A., S.G., A.K. contributed to critical review of manuscript.

## Competing interests

The authors declare no competing interests.

## Additional information

[1]Department of Epidemiology, Erasmus University Medical Center, Rotterdam, the Netherlands. [2]Department of Genetic Identification, Erasmus University Medical Center, Rotterdam, the Netherlands. [3]Epidemiology and Microbial Genomics, National Health Laboratory, Dudelange, Luxembourg. [4]Department of Gastroenterology and Hepatology, Erasmus University Medical Center, Rotterdam, the Netherlands. [5]Friedman

School of Nutrition Science and Policy, Tufts University, Boston, MA, USA. [6]Population Sciences Branch, National Heart, Lung, and Blood Institute, National Institutes of Health, Bethesda, Maryland and the Framingham Heart Study, Framingham, MA, USA. [7]Institute of Molecular Medicine, McGovern Medical School, University of Texas Health Science Center at Houston, Houston, TX, USA. [8]Department of Epidemiology, Johns Hopkins Bloomberg School of Public Health, Baltimore, MD, USA. [9]Research Unit of Molecular Epidemiology, Helmholtz Zentrum München, German Research Center for Environmental Health, Neuherberg, Germany. [10]Institute of Epidemiology, Helmholtz Zentrum München, German Research Center for Environmental Health, Neuherberg, Germany. [11]Division of Clinical Epidemiology and Aging Research, German Cancer Research Center (DKFZ), Heidelberg, Germany. [12]MRC Integrative Epidemiology Unit, Population Health Sciences, Bristol Medical School, University of Bristol, Bristol, UK. [13]AMCHSS, Sree Chitra Tirunal Institute for Medical Sciences and Technology, Thiruvananthapuram, Kerala, India. [14]Epigenetics Group, International Agency for Research on Cancer (IARC), Lyon Cedex 08, France. [15]Laboratory of Biostatistics, Department of Biomedical Sciences, University of Sassari, Sassari, Italy. [16]MRC Centre for Environment and Health, Department of Epidemiology and Biostatistics, School of Public Health, St Mary's Campus, Imperial College London, Norfolk Place, London, UK. [17]UK Dementia Research Institute at Imperial College London, London, UK. [18]Imperial College NIHR Biomedical Research Centre, London, UK. [19]Department of Twin Research and Genetic Epidemiology, Kings College London, London, UK. [20]Epigenetics Programme, Babraham Institute, Cambridge, UK. [21]Cardiovascular Health Research Unit, Department of Medicine, University of Washington, Seattle, CHRU, Seattle, WA, USA. [22]Department of Genetics, University Medical Center Groningen, University of Groningen, Groningen, The Netherlands. [23]Epidemiology and Prevention Unit, IRCCS National Cancer Institute Foundation, Milan, Italy. [24]Italian Institute for Genomic Medicine (IIGM, former HuGeF), c/o IRCCS Candiolo, Candiolo, Italy. [25]Department of Psychiatry, Amsterdam UMC, Amsterdam, the Netherlands. [26]Department of Biomedical Sciences, Chang Gung University, Taoyuan, Taiwan. [27]Genomic Medicine Research Core Laboratory, Chang Gung Memorial Hospital, Linkou, Taiwan. [28]Nutritional Epidemiology Group, International Agency for Research on Cancer, Lyon, France. [29]Genomics plc, Park End St, Oxford, UK. [30]Oncode Institute, Utrecht, The Netherlands. [31]Department of Internal Medicine, Erasmus University Medical Center, Rotterdam, The Netherlands. [32]German Center for Cardiovascular Research (DZHK), Partner Site Munich Heart Alliance, Munich, Germany. [33]Computational Medicine Core at Center for Lung Biology, Division of Pulmonary, Critical Care and Sleep Medicine, Department of Medicine, University of Washington, Seattle, WA, USA. [34]Health Data Research UK-London, London, UK. [35]Division of Preventive Oncology, German Cancer Research Center (DKFZ) and National Center for Tumor Diseases (NCT), Heidelberg, Germany. [36]German Cancer Consortium (DKTK), German Cancer Research Center (DKFZ), Heidelberg, Germany. [37]Network Aging Research, University of Heidelberg, Heidelberg, Germany. [38]Department of Genetics, School of Medicine, Mashhad University of Medical Sciences, Mashhad, Iran. ✉email: m.ghanbari@erasmusmc.nl

