## [Peer Review File · Nature Communications]

REVIEWER COMMENTS

Reviewer #1 (Remarks to the Author):

The paper Karabegovic et al describes the largest EWAS for coffee and tea consumption, i.e. association between DNA-methylation at around 400,000 CpG-sites and coffee or tea consumption. They also try to infer causality using MR.

The paper is well written, easy to follow and presents some interesting novel results. However, I have some major and minor comments:

- 1) This study was done for both coffee and tea. 11 CpG-sites were found to be novel for coffee and 2 for tea. However, it feels like you have forgotten to write about the tea part, both throughout the methods section, results and discussion. For example, follow up studies is only presented for coffee and the first section of discussion is only presenting the coffee results.
- 2) A clearer description about overlap between identified CpG-sites and previous coffee and tea GWASes needs to be added. preferable in a table. So the reader can compare all hits with position.
- 3) you present that some CpG-sites are replicated by p-value threshold, but you don't mention if the direction is in the same direction.
- 4) The AHRR gene is, as the authors have acknowledged, previously been both associated with coffee in previous GWAS and with smoking in previous EWAS.
 - a) Since you don't have a clear comparison with GWAS results, it is not possible to identify if your hits are true novel coffee hits, or just novel EWAS coffee hits. For example AHRR, you can't determine if this association is driven by DNA-methylation or GWAS SNPs? It is not clear if you adjusted for underlying SNPs in these regions that might actually drive the association rather than DNA methylation
 - b) Since AHRR is strongly associated with smoking and coffee consumption also are highly correlated with smoking (coffee drinkers tend to smoke more than non-coffee drinkers and vice versa for tea). I think you should stratify the analysis, only analysing non-smokers.
- 5) Why not stratify for sex and analyse males and females separately? Could find interesting effects
- 6) I also think it would make sense to adjust the coffee analysis for tea consumption, and the tea analysis for coffee consumption
- 7) The 11 significant CpGs (in the meta-analysis) were annotated to 9 genes. I would like to see a better discussion around previous genes identified for coffee (and for tea). What was found in this study and not?
- 8) For the discovery cohort, you used a suggestive threshold of 5×10^{-6} . It is not clear how this threshold was set.
- 9) the two first sections in discussion both start with presenting 11 hits. Please remove one sentence. Also add tea results.
- 10) It might be a problem with the pdf, but the text below the Manhattan plot is missing. Should clearly explain the figure. And there are 4 different tables without headers (might belong to Table 1).

Reviewer #2 (Remarks to the Author):

Authors conducted a meta-analysis of epigenome-wide association studies (EWAS) of DNA

methylation for coffee and tea consumption dosages. This study enrolled in total ~16,000 individuals of European and African-American ancestry. The study consisted of both discovery and replication stages with the large number of the participants, which complements relatively less reproducible results of methylation EWAS. Mendelian randomization (MR) study did not find causal inference of coffee consumption-associated variants on coffee-associated methylation. Authors then focused of the expression regulatory profiles of the identified methylation site on PHGDH expression. This is one of the well-conducted and large-scale methylation EWAS on human dietary habits.

1. Genome studies on human habits should enhance their values when a set of the habits are parallelly assessed. While authors focused on coffee and tea consumptions, since there exist data on other habits or traits related to methylation (i.e., alcohol drinking, smoking, obesity, and others), similar EWAS studies should also be done for these traits in this manuscript.
2. Assessing multiple human habits can also provide interpretation of the MR analysis. For example, we can relatively know whether lack of MR link in coffee consumption was due to lack of power or not.
3. Readers may want to know why the EWAS results for tea consumption lacked significant hits.
4. How about the MR analysis for tea consumption (and other habits as well)?
5. While food frequently questionnaires (FFQ) is widely adopted and may be the most appropriate measurement in these types of study, potential limitation and bias due to FFQ should be discussed.

Reviewer #3 (Remarks to the Author):

The report "Epigenome-wide association meta-analysis ..." by Karabegovic et al. examines relationships between coffee or tea consumption and methylation across the genome in a large total study population assembled through meta-analysis of cohort level analysis, including individuals of European and African ancestry. The authors identify several loci meeting experiment-wide significance thresholds for coffee but not tea, some with quite plausible connections to known molecular effects /pathways related to coffee consumption, giving the study strong credibility, e.g. AHRR gene. One association is at the PHGDH gene has been implicated in other liver functions and follow-up analysis characterizes and extends a potential role of this gene. The findings are highly plausible and interesting, and they offer insights into candidate mechanisms for the physiologic effects of coffee consumption and potentially on health outcomes, although additional research is needed.

Specific comments

1. Beginning in the abstract and continuing throughout, the authors advance the notion of "contradictory findings" for associations with coffee and tea with health outcomes. It might be more accurate to balance this idea with the highly consistent associations of coffee intake and reduced T2D. See for example lines 110-115.
2. Abstract, line 92, phrase "lipid-associated genes" is a little vague. More accurate would be genes associated with circulating lipids.
3. The abstract is confusing on the point of separate scans for coffee and tea: "... EAS on coffee and tea consumption ..." (line 85). The scans were separate and there is no mention of the null result with tea in the abstract. See also line 126.
4. Introduction, lines 99-100. The introduction talks about how preference for coffee v. tea varies. What is written is true, but much of the variation pertains to the interindividual quantity of coffee and/or tea consumed rather than the choice of coffee v. tea v. both v. neither, which can be

influenced by both culture and personal preference.

5. Line 110 – Please check whether LDLC increasing effects are related to whether coffee is filtered through paper or not.

6. There are some moderately important issues with the formulation of the outcome variable (i.e. coffee or tea) and the statistical model for hypothesis testing.

a. The authors do not appear to distinguish drinkers from non-drinkers, applying instead a linear model to quantity of consumed coffee/tea. They should provide information about the fraction not drinking at all (apologies if I missed this). If this fraction is minimal, there isn't a problem. However, if a substantial proportion of the study population abstains, then it may be important to consider analysis restricted to drinkers, or comparing drinkers to non-drinkers, e.g. with a logistic model.

b. Do the authors think that coffee drinking confounds analysis for tea, and vice versa? Should this be addressed in the analysis or at least in the Discussion? How do the authors account for individuals who drink both coffee and tea? It would not be surprising if there are individuals who drink very little tea but a large amount of coffee. What would the authors expect of their methylation signal compared to individuals who drink a lot of tea? This thread may be relevant to the discussion starting on line 490.

c. The authors may consider trying to address unique effects of coffee or tea, e.g. by restricting to strata of drinkers exclusively of one or the other.

d. Is there any evidence of association, e.g. nominal significance, with tea consumption at loci related to coffee consumption or with coffee consumption at loci (suggestively) related to tea consumption?

e. Could various sensitivity analyses implied by the preceding be performed? The authors could limit this analysis to the largest contributing cohort, the FHS, to reduce workload.

7. These considerations are partly related to the issue of whether caffeine or some other component of coffee is driving the associations. It's not critical but the authors could consider looking at caffeine from all sources, i.e. combining tea, coffee, caffeinated beverages, and ask whether the effect they observe are stronger than for coffee alone. This analysis, for example, might be done in just the largest cohort (i.e. the FHS).

8. It's not totally clear why model 2 is labeled the "most conservative model." Including covariates always runs a risk of inducing associations, e.g. by collider bias, even if it may also account for known confounding.

9. What is the basis for thinking, a priori, that methylation patterns will be different in EA and AA, justifying stratified analysis (lines 211 & 388)? Are coffee/tea drinking habits very different in the two ancestries? One might expect that the same genes would be involved in both ancestry groups and the sequences around the genes should be essentially identical. Perhaps they could discuss nominal evidence for reciprocal associations, e.g. EA top loci in AA, and vice versa (Table S6)? Can the authors cite evidence for well-powered lack of trans-ancestry replication of methylation sites in other studies? If so, what are the biological mechanisms that result in (validated) methylation differences between EA and AA.

10. The forest plots from each significant locus do not show the overall estimate. Please provide. It's also clear from the presentation that the effects are dominated by the FHS contribution. Would the authors consider sensitivity analysis that removes the FHS contribution, or examines the FHS association alone?

11. Table 2 should show standard errors for the beta estimates.

12. The AHRR gene is a great candidate for effects of coffee. However, the authors note that the top CpGs are eQTM's not for AHRR but for the neighboring gene EXOC3. Do they think there may be an eQTM for AHRR in a different tissue, e.g. liver, as is apparently demonstrated for the CpG near PHGDH in ref. 54.

13. For a null MR results, such as here, one typically would not perform sensitivity analysis as the

authors did. It's not totally clear how a significant result in the sensitivity analysis would be reconciled with null primary analysis. Yes, there could be an argument that primary effects are confounded toward the null due to pleiotropy. However, if the methods that try to account for pleiotropy were to yield a significant result, there may still remain questions about whether the finding may still be driven by pleiotropy that is not addressed by the sensitivity methods. In the MR, there is also no discussion of how genetic effects on methylation may vary by tissue, even that the cis-meQTL data derives from blood and other tissues may be relevant, e.g. liver (Methods, and Discussion starting line 465).

14. In spite of the authors claims to the contrary, causality in the conclusions is in doubt (p. 18). The MR was negative but this result is attributed to lack of power, even in the absence of a power calculation, which should be provided. The authors cite additional reasons for a null MR result. However, not mentioned is the alternative conclusion that the methylation patterns reflect pathways of coffee consumption effects on the body but are not causal. It's not completely clear what the arrows imply in figure S2. However, to the extent they represent causality, the figure is incomplete in not considering reverse directions, e.g. from liver function to methylation, or from coffee consumption to methylation, or even from expression to methylation, the latter, e.g. in trans.

15. The comments about smoking are interesting. The authors note that smoking is associated with methylation at some of the same sites. They also note that smoking and coffee drinking are correlated, but even after adjusting for smoking status, the associations between coffee consumption and methylation at these sites with coffee consumption persists. Are the effects at these sites associated with smoking in the same direction as the effects associated with coffee drinking? Please comment/address.

16. Please refine the discussion of HDAC4. The authors note its role in xenobiotic metabolism but its molecular function is as a histone deacetylase. Is xenobiotic metabolism the only pathway in which it is active?

17. How do the authors justify genome-wide threshold in this experiment? Is there a reference for the threshold of $p < 10^{-7}$ in this kind of methylation scan?

We are deeply grateful to the reviewers for taking the time to provide us with constructive comments and valuable suggestions. Please find below a point-by-point response to each of the comments. We indicate revisions in the updated manuscript by a yellow highlighter together with line numbers.

Reviewer #1:

The paper is well written, easy to follow and presents some interesting novel results. However, I have some major and minor comments:

R1-Q1. This study was done for both coffee and tea. 11 CpG-sites were found to be novel for coffee and 2 for tea. However, it feels like you have forgotten to write about the tea part, both throughout the methods section, results and discussion. For example, follow up studies is only presented for coffee and the first section of discussion is only presenting the coffee results.

Response. We thank the reviewer for pointing out the unclarity in the results of tea part, which we would gladly elaborate on these here. The EWAS meta-analysis of tea consumption revealed two CpGs suggestively associated ($P\text{-value} < 5.0 \times 10^{-6}$). As no CpG had surpassed the EWAS significance threshold of 1.1×10^{-7} for tea association, the follow up analyses were previously presented only for the 11 CpGs significantly associated with coffee consumption. However, we agree with the reviewer that the follow up studies should be done for the two suggestively tea-associated CpGs as well. Thus, we have now incorporated into the manuscript the additional analyses including the assessment of expression quantitative trait methylation, methylation quantitative trait loci, and gene differential expression analysis for the tea-associated CpGs. Regarding the Mendelian randomization (MR) analysis, it is important to note that the literature on GWAS related to tea consumption is quite limited. To the best of our knowledge, there is just one large-scale GWAS on tea consumption [*GWAS of 165,084 Japanese individuals identified nine loci associated with dietary habits. Matoba et al., Nature human behaviour. 2020*] (1). This study reported only one significant SNP; however, the same SNP is also associated with multiple other dietary habits (e.g., milk, yogurt, coffee, tofu) in the Japanese population. Considering that our EWAS meta-analysis of tea consumption did not yield epigenome-wide significant hits, and that there is no valid genetic instruments for tea consumption (only one GWAS SNP from Japanese population that is associated with multiple dietary habits), we opted not to perform the MR analysis for tea consumption. The manuscript has now modified accordingly to make it more readable.

a. Abstract:

Page 3, line 94-95: “EWAS meta-analysis on tea consumption showed only two CpGs annotated to CACNA1A and PRDM16 that were suggestively associated ($P\text{-value} < 5.0 \times 10^{-6}$).”

b. Results:

Page 15, line 408-409: “The tea-suggestive associated genes showed to be down-regulated in breast tissue.”

Page 15, line 412-414: “Pathway analysis of the two genes suggestively associated with tea consumption showed enrichment for white Adipose tissue browning ($P=3.2\times 10^{-5}$), nNOS signalling in skeletal muscle cells ($P=3.6\times 10^{-3}$), and Maturity onset diabetes of young (MODY) ($P=5.2\times 10^{-3}$) (Table S13).”

Page 15, line 415-418: “The coffee-associated CpGs as well as two CpGs suggestively associated with tea were further explored in association with genetic variations (meQTL) or expression levels of their nearby or distant genes (eQTM) using the BIOS-BBMRI database. Eight of the coffee CpGs and one of the tea CpGs (cg20099906) were associated with genetic variants in the neighbouring genes (cis-meQTLs) (Table S14).”

Page 16, line 427: “We did not find eQTM associated with the two suggestive CpGs for tea consumption.”

c. Discussion:

Page 18, line 475-476: “Additionally, we found two CpGs that were suggestively associated with tea consumption.”

Page 22, line 596-597: “Among the CpGs suggestively associated with tea consumption, cg20099906 is annotated to *CACNA1A* and cg05804170 annotated to *PRDM16*.”

Page 22, line 601-605: “Moreover, *PRDM16* gene encodes a zinc finger transcription factor which controls the development of brown adipocytes in brown adipose tissue (2). Alteration of *PRDM16* expression has been linked to several outcomes including cardiomyopathy, obesity, metabolic syndrome and hypertension (3, 4). Future studies with even larger sample sizes might be needed to replicate the suggestive associations with tea intake.”

R1-Q2. A clearer description about overlap between identified CpG-sites and previous coffee and tea GWASes needs to be added, preferable in a table. So the reader can compare all hits with position

Response. We agree with the reviewer that it would be more comprehensible to have a clear overview on the overlap between the newly identified CpGs (loci) associated with coffee consumption and previously reported genes associated with coffee and tea by GWAS. We have curated information from both GWAS catalog (<https://www.ebi.ac.uk/gwas/>) as well as Google Scholar about GWAS of coffee and tea consumption. In addition, we have included two previous EWAS on either coffee and tea consumption in supplementary Table S5. The description and inclusion of the table have been added to the Results section as follow:

Page 13, line 354-360: “In order to determine the novelty of our findings, we examined whether the loci identified for coffee and tea consumption have been found

previously by GWAS or other EWAS. None of the loci had been reported before to be associated with either coffee or tea consumption (Table S5). In addition, the association of cis-meQTLs of coffee-associated CpGs were looked up in publicly available GWAS of coffee intake, published with UK-Biobank data (n= 358,093) and available through GWAS ATLAS (<https://atlas.ctglab.nl/>). We did not find any of the lead SNPs of CpGs-meQTLs to be associated with coffee intake (Table S6).”

R1-Q3. You present that some CpG-sites are replicated by p-value threshold, but you don't mention if the direction is in the same direction.

Response. In Table 2A, the beta estimates (β) of the coffee-associated CpGs do provide the direction of effect in the discovery and replication panels.

R1-Q4. Since you don't have a clear comparison with GWAS results, it is not possible to identify if your hits are true novel coffee hits, or just novel EWAS coffee hits. For example *AHRR*, you can't determine if this association is driven by DNA-methylation or GWAS SNPs? It is not clear if you adjusted for underlying SNPs in these regions that might actually drive the association rather than DNA methylation.

Response. We thank the reviewer for this comment and the opportunity to clarify this part. As mentioned above in response to R1-Q1, we have now integrated information from both GWAS catalog and Google Scholar to provide a comparison with the GWAS results. In addition, we have included previous EWAS on coffee and tea consumption in Supplementary Table S5. Furthermore, we considered important to examine the association between meQTLs of each coffee-CpG with coffee consumption.

Regarding the possibility of SNP-driven association, it is worth mentioning that some probes (CpGs) have been excluded from the EWAS output prior meta-analysis. The criteria used for excluding these CpGs were CpGs with a SNP in the same genomic region, non-CpG probes, and cross-reactive probes. Therefore, we have already ruled out the possibility that SNPs drive the observed associations. This information has now added to the Method section. We have also clarified this issue in the Results section.

About the *AHRR* gene, it seems that our discussion might have been unclear for the reviewer, maybe because of the lack of a clear comparison with GWAS results. In fact, *AHR* along with *CYP1A1* and *CYP2A2* genes have been associated with coffee consumption in previous GWAS studies (PMID: 25288136, 21357676, 27702941, and 31837886) (Table S5). These genes belong to the Xenobiotic pathway. In the EWAS meta-analysis of coffee consumption, four of the identified CpGs are annotated to the Aryl Hydrocarbon Receptor Repressor (*AHRR*) gene, which is a repressor of the *AHR* gene. Hence, we speculated that the methylation within *AHRR* might influence the Xenobiotic pathway via functionally related genes (*AHR*, *CYP1A1* and *CYP2A2*). We have now revised the Discussion to make it more readable.

a. Methods:

Page 7, line 204-206.: “Prior to their inclusion into the meta-analysis, all probes with a SNP, non-CpG probes and cross-reactive probes were removed as suggested by Chen *et al.* (5)”.

b. Results:

Page 13, line 354-360: “In order to determine the novelty of our findings, we examined whether the loci identified for coffee and tea consumption have been found previously by GWAS or other EWAS. None of the loci had been previously reported to be associated with either coffee or tea consumption (Table S5). In addition, the association of *cis*-meQTLs of coffee-associated CpGs were looked up in publicly available GWAS data of coffee intake, published with UK-Biobank data (n= 358,093) (6) and available through GWAS ATLAS (<https://atlas.ctglab.nl/>). We did not find any of the lead SNPs of CpGs-meQTLs to be associated with coffee intake (Table S6).”

c. Discussion:

Page 23, line 606-610: “None of the genes reported in this study have been previously associated with coffee or tea consumptions. However, *AHRR* is a repressor of the *AHR* gene. Almost all of the previous GWAS and EWAS conducted on coffee intake, or its related traits, have described the critical role of *AHR* and Cytochrome P450 (*CYP1A1* and *CYP1A2*) on pharmacokinetic and pharmacodynamic properties of the caffeine component of coffee underlying a genetic propensity to consume the beverage (7-9).”

R1-Q5. Since *AHRR* is strongly associated with smoking and coffee consumption also are highly correlated with smoking (coffee drinkers tend to smoke more than non-coffee drinkers and vice versa for tea). I think you should stratify the analysis, only analysing non-smokers.

Response. We acknowledge the reviewer’s suggestion. We have now performed stratified analysis on non-smokers in the largest contributing cohort, Framingham Heart Study (FHS), and the Rotterdam Study (RS), as kindly suggested by Reviewer #3. Further information is detailed in Methods section and the results of stratified analysis are provided in Table S9.

a. Methods

Page 8, line 211-215: “In addition, we stratified two of the largest cohort (FHS and RS) by sex, smoking status and coffee consumption frequency. Smoking status was evaluated on current vs never smokers; coffee consumption frequency was determined as none or infrequent drinkers (<1 cup/day), moderate drinkers (>=1 and <4 cup/day), and high or frequent drinkers (>=4 cup/day) using a large-scale GWAS on coffee consumption as a reference for categorization of the variable (7).

b. Results

Page 14, line 370-377: “To minimize the possible confounding effect of smoking on the association between coffee consumption and DNA methylation, we also conducted EWAS on participants who self-reported as never smokers (n=2,123). The results of these analyses are shown in Supplementary Table S9. When analysing current and former smokers, the effect sizes did not change substantially compared to the overall sample. In contrast, the results from the subset of never-smokers indicated that the effect sizes of six out of the eleven lead probes namely:

cg09935388, cg21161138, cg25648203, cg06126421, cg05575921, and cg03636183, were increased comparing to the beta estimates of the meta-analyzed sample.”

c. Discussion

Page 18, line 492-495: “When we restricted our analysis to never smokers, the effect sizes of six out of the eleven lead probes were increased. The differences in the effect sizes and the overlap of some of our CpGs with findings from previous large-scale EWAS on smoking may indicate that our results could be affected by some residual confounding of smoking.”

R1-Q6. Why not stratify for sex and analyse males and females separately? Could find interesting effects.

Response. Following the same reasoning as mentioned above, we have now performed stratified analysis on sex in the Framingham Heart Study (FHS) and the Rotterdam Study (RS). The results of sex stratified analysis are reported in Table S9 and the manuscript has been modified accordingly.

a. Methods

Page 8, line 211-212: “In addition, we stratified two of the largest cohort (FHS and RS) by sex, smoking status and coffee consumption frequency.”

b. Results

Page 15, line 396-399: “Lastly, when assessing males and females separately, five CpGs were nominally significant among males and four among females that could be attributed to power; as the effect sizes remained similar compared to the overall sample (Table S9).”

R1-Q7. I also think it would make sense to adjust the coffee analysis for tea consumption, and the tea analysis for coffee consumption.

Response. We acknowledge the reviewer’s comment and the relevance of addressing the potential confounding effect of tea or coffee on the other beverage. We have now included tea and coffee consumption in the EWAS-regression models of the other beverage. A description of these analyses were added to Method section and the findings were presented in Table S9. These sensitivity analyses were also performed in the FHS and RS.

a. Methods

Page 7, line 195-197: “Tea and coffee consumption were also added as covariates to Model 2 of EWAS on coffee and tea, respectively, in order to assess potential confounding effects on DNA methylation. This analysis was conducted on FHS, largest cohorts, and RS”.

b. Results

Page 14, line 386-391: “When adjusting by tea consumption, the effect estimates of the majority (eight CpGs) of the identified methylation sites, did not change substantially. We observed a similar pattern (same direction in the effect estimate) for the rest of the CpGs that had somewhat change in the effect estimate compared to the main model (Table S9). For example, our main model showed a $\beta = -0.001$ for cg21161138 ($P = 6.66 \times 10^{-12}$), while the additional adjustment for tea consumption in FHS and RS showed a smaller effect ($\beta = -0.0003$).”

R1-Q8. The 11 significant CpGs (in the meta-analysis) were annotated to 9 genes. I would like to see a better discussion around previous genes identified for coffee (and for tea). What was found in this study and not?

Response. We have now incorporated genes identified by previous GWASs of coffee in supplementary Table S5, which makes it possible to compare them with our novel hits by EWAS (Table 2A). Notably, we did not find an overlap between the significant genes (loci) in our EWAS and findings of previously published GWAS or EWAS of coffee-consumption. In the Discussion part, we have indicated that four of the significant CpGs are annotated to the aryl hydrocarbon receptor repressor (*AHRR*), which is a repressor of *AHR*. The *AHR* gene has been previously reported in multiple coffee consumption GWAS (PMID: 25288136, 21357676, 29367735, 31345160, 31959922, 31837886, and 27702941). We have now incorporated the remarks of the Reviewer in the Results and Discussion sections.

a. Results

Page 13, line 354-356: “In order to determine the novelty of our findings, we examined whether the loci identified for coffee and tea consumption have been found previously reported by GWAS or other EWAS. None of the loci had been previously reported to be associated with either coffee or tea consumption (Table S5).”

b. Discussion

Page 23, line 606-617: “None of the genes reported in this study have been previously associated with coffee or tea consumptions. However, *AHRR* is a repressor of the *AHR* gene. Most of the previous GWAS and EWAS conducted on coffee intake, or its related traits, have described the critical role of *AHR* and Cytochrome P450 (*CYP1A1* and *CYP1A2*) on pharmacokinetic and pharmacodynamic properties of the caffeine component of coffee underlying a genetic propensity to consume the beverage (7-9). Literature on other coffee-associated genes including *GCKR*, *ABCG2*, *SORCS2*, among others, has shown an association with smoking initiation, higher adiposity, fasting insulin and glucose, but lower blood pressure and favourable lipid, inflammatory as well as liver enzyme profiles (7). Regarding tea consumption, previous evidence have determined the importance of *ALDH2*, *DNAJC16*, and *TTC17* genes in tea metabolism and have described their role in cancer-related pathways and

interaction with estradiol in female population (1, 10). However, we did not observe any significant association between CpGs annotated to these genes and tea consumption in our study.”

R1-Q9. For the discovery cohort, you used a suggestive threshold of 5×10^{-6} . It is not clear how this threshold was set.

Response. We appreciate the reviewer’s comment and the opportunity to further clarify this. Most of the cohorts included in this meta-analysis have used the Illumina 450k Infinium methylation beadchip which interrogates ~450,000 methylation sites (CpGs) per sample at single nucleotide resolution. For testing 450K CpG, a (stringent) Bonferroni corrected p-value threshold of 1.1×10^{-7} has been widely used as the genome-wide significance in the majority of previous EWASs (11-14). However, it has been shown that there are co-methylation regions (DMRs) in this methylation array and some previous EWASs have suggested FDR-based thresholds of 1% to 5% as the significance thresholds, which correspond to *P*-values close to $P = 1 \times 10^{-4}$ (15, 16). In the discovery phase of our EWAS on coffee and tea consumption, we therefore considered a suggestive threshold (5.0×10^{-6}), between these thresholds suggested by previous studies, not to miss any important signal because of a lack of power. In the EWAS meta-analysis combining overall samples of the discovery and replication phases, we then considered CpGs surpassing the Bonferroni adjusted genome-wide significance *P*-value threshold of 1.1×10^{-7} as significant. We now have modified the text to make this point clear in the manuscript.

R1-Q10. The two first sections in discussion both start with presenting 11 hits. Please remove one sentence. Also add tea results.

Response. We thank the reviewer for attentively reading the manuscript. We have modified this part accordingly.

R1-Q11. It might be a problem with the pdf, but the text below the Manhattan plot is missing. Should clearly explain the figure. And there are 4 different tables without headers (might belong to Table 1)

Response. We have adapted the legend of Manhattan plot and headers of tables accordingly.

Figure 2. “Manhattan plots depict the results of epigenome-wide association studies (EWAS) in overall sample size for coffee (A) and tea (B) consumption in the fully adjusted models. Each dot corresponds to a single CpG site plotted as the negative logarithm of the p-value ($-\log(\text{p-value})$) (y-axis) against the genomic position of the CpG site (x-axis). The red line indicates the Bonferroni adjusted genome-wide significance *P*-value threshold of 1.1×10^{-7} .”

Reviewer #2:

This is one of the well-conducted and large-scale methylation EWAS on human dietary habits.

R2-Q1. Genome studies on human habits should enhance their values when a set of the habits are parallelly assessed. While authors focused on coffee and tea consumptions, since there exist data on other habits or traits related to methylation (i.e., alcohol drinking, smoking, obesity, and others), similar EWAS studies should also be done for these traits in this manuscript.

Response. We acknowledge the reviewer's comment. There exists data on other habits or traits related to DNA methylation in some of the cohorts participating in this study; however, similar large-scale EWAS studies have already been published on these traits where the cohorts mentioned have participated with the data. For example, EWAS meta-analysis of alcohol drinking was performed prior within the CHARGE consortium (n= 13,317) (17), or EWAS meta-analysis of smoking (n= 15,907) (18) within the CHARGE consortium, and also EWAS of obesity-related traits (n= 3,547) within the Rotterdam Study and ARIC cohorts (19). Furthermore, the research is increasing in not that commonly investigated traits or habits that might be associated with DNA methylation such as cocaine and crack (20) and cannabis exposure (21). While we agree with the reviewer that other habits could also modify epigenetic patterns, we are concerned that the topic of the manuscript would be too broad to run EWAS on all these traits together and that would not bring much novelty given the previously EWAS reported findings. Another concern is heterogeneity, combining all these traits together may contribute to increase the heterogeneity in the association results. Alternatively, we have looked up in the online tools of EWAS findings (EWAS Atlas and EWAS Catalog) to find whether the coffee- or tea-associated CpGs have been reported for the other traits in previous EWAS studies. We have provided a more detailed overview of our newly identified CpGs for coffee/tea that are linked to other traits and human habits in Supplementary Table S15. We have also integrated an upset plot to provide an overlap of the CpGs with some traits of the previously mentioned EWASs (Figure S8).

R2-Q2. Assessing multiple human habits can also provide interpretation of the MR analysis. For example, we can relatively know whether lack of MR link in coffee consumption was due to lack of power or not.

Response. We thank the reviewer for this remark. Indeed, sample size might be a strong determinant for the non-significant causality results between coffee consumption and DNA methylation. We are concerned that the inclusion of additional traits will be subject to multiple testing correction and will influence the p-value significance level for causality, which is already affected by the lack of power. In addition, as mentioned in response to the previous comment, combining different traits can increase heterogeneity and may makes the interpretation of results more difficult. Regarding the lack of power in our MR studies, it is worth mentioning that a recent review of MR studies of coffee and caffeine consumption indicates that substantial sample size is needed to assess the causality (22). The review shows that the majority of the studies reported non-significant findings and stated underpowered IV as a limitation despite large sample sizes (in some cases $N \geq 100,000$). Thus, extensive sample sizes might be needed in order to provide substantial power and draw causal inference on coffee consumption (and other habits). For more information please also see our response to R3-Q18. We have now modified the Discussion in order to make it more clear.

Page 20, line 543-552: “the power of our MR analyses was limited apparently due to the small sample size of our DNA methylation data, which is evident in small effect sizes and R^2 (variance explained) values. The lack of strong genetic instruments for coffee exposure might also be an important determinant of the results observed. In line with this reasoning, a recent review of MR studies of coffee and caffeine consumption (22) has indicated that substantial sample size is indeed needed to assess the causality when it comes to coffee and other dietary habits. The majority of the studies included in this review reported non-significant findings and stated underpowered instrumental variables as a limitation despite large sample sizes (in some cases $N \geq 100,000$). Therefore, for MR studies on coffee consumption, extensive sample sizes might be needed in order to provide substantial power and infer causality.”

R2-Q3. Readers may want to know why the EWAS results for tea consumption lacked significant hits.

Response. We acknowledge the reviewer’s comment. We have now elaborated on the potential reasons that EWAS results for tea consumption lacked significant hits in this study.

Discussion.

Page 22, line 585-595: “Therefore, specific nutrients that are present only in coffee (or present in higher abundance compared to tea) such as caffeine or various phenolic compounds (23) might be a plausible explanation resulting in stronger effect evident in significant association with DNA methylation compared to tea consumption. Moreover, the data collection on tea consumption might result in more heterogeneity (e.g., due to different types of tea) compared to coffee. It is also worth mentioning

that previous literature indicated many GWASs on coffee consumption (Table S5), but only one GWAS of tea consumption in Japanese population (1). We cannot deduce with certainty the reasoning behind this, but one might suspect the weaker effect of tea consumption and potential existence of publication bias due to the lack of non-significant findings. In the future, perhaps bigger sample size would provide more understanding in tea consumption EWAS. However, given the well-powered EWAS conducted here it is also possible that tea consumption has no discernible impact on peripheral blood DNA methylation.”

R2-Q4. How about the MR analysis for tea consumption (and other habits as well)?

Response. Regarding the MR analysis for tea consumption, we should note that the literature on GWAS with tea consumption is limited. To the best of our knowledge, there is only one large-scale GWAS on tea consumption [GWAS of 165,084 Japanese individuals identified nine loci associated with dietary habits. Matoba et al., *Nature human behaviour*. 2020] (1). Based on their findings, there is only one significant SNP that could be used as an instrumental variable for the MR analysis; however, the same SNP is associated with multiple dietary habits (incl. milk, yogurt, coffee, tofu) in the same Japanese population. Considering that our EWAS meta-analysis of tea consumption did not yield epigenome-wide significant hits (only two CpGs suggestively associated) and that there was no valid genetic instruments for tea consumption (only one GWAS SNP from Japanese population that is also associated with multiple dietary habits), we opted not to perform the MR analysis for tea consumption. For other habits (e.g., alcohol drinking, smoking, obesity), the MR analysis have been performed and published in previous large-scale EWAS studies (24-26).

R2-Q5. While food frequently questionnaires (FFQ) is widely adopted and may be the most appropriate measurement in these types of study, potential limitation and bias due to FFQ should be discussed.

Response. We thank the reviewer for pointing out this limitation of our study. We have now incorporated the following paragraph in the Discussion section.

Discussion

Page 24, line 646-651: “It is also important to address the potential limitation of this study resulting from the method of dietary data collection via FFQ. While the FFQ are the most cost-effective way to collect the dietary data in cohort studies, they are subject to some inaccuracies (e.g. participants might have a recall bias when filling in the questionnaires). Nevertheless, when compared to the other dietary assessment methods (e.g. 24-h recall, food diary, etc.), the FFQ usually reports a lower within-person variation (27).”

Reviewer #3:

The findings are highly plausible and interesting, and they offer insights into candidate mechanisms for the physiologic effects of coffee consumption and potentially on health outcomes, although additional research is needed.

Specific comments:

R3-Q1. Beginning in the abstract and continuing throughout, the authors advance the notion of “contradictory findings” for associations with coffee and tea with health outcomes. It might be more accurate to balance this idea with the highly consistent associations of coffee intake and reduced T2D. See for example lines 110-115.

Response. We acknowledge the reviewer’s comment. We have modified this part as suggested in the Abstract.

Page 3, line 82-85: “Coffee and tea are extensively consumed beverages worldwide and have received considerable attention regarding health risks and benefits. In observational studies, these beverages have been consistently associated with reduced risk of type 2 diabetes and liver diseases; however, their effects on other health outcomes remain controversial.”

R3-Q2. Abstract, line 92, phrase “lipid-associated genes” is a little vague. More accurate would be genes associated with circulating lipids.

Response. We have revised the phrase in the Abstract accordingly.

Page 3, line 91-94: “Knockdown of *PHGDH* expression in liver cells showed a correlation with expression levels of genes associated with circulating lipids, suggesting a role of *PHGDH* in hepatic-lipid metabolism.”

R3-Q3. The abstract is confusing on the point of separate scans for coffee and tea: “... EAS on coffee and tea consumption ...” (line 85). The scans were separate and there is no mention of the null result with tea in the abstract. See also line 126.

Response. We have now modified the Abstract as follow:

Page 3, line 94-95: “EWAS meta-analysis on tea consumption showed only two CpGs annotated to *CACNA1A* and *PRDM16* that were suggestively associated (P-value < 5.0×10^{-6}).”

R3-Q4. Introduction, lines 99-100. The introduction talks about how preference for coffee v. tea varies. What is written is true, but much of the variation pertains to the interindividual quantity of coffee and/or tea consumed rather than the choice of coffee v. tea v. both v. neither, which can be influenced by both culture and personal preference.

Response. We thank the reviewer for carefully reading the manuscript. We have revised the Introduction following the reviewer's suggestion.

Page 4, line 102-105: "The preference for one or the other and the quantity of coffee and tea consumed vary between individuals (28), which can be influenced by geographical region as well as both cultural and personal preference."

R3-Q5. Line 110 – Please check whether LDLC increasing effects are related to whether coffee is filtered through paper or not.

Response. We thank the reviewer for this remark. We have included the statement below about the lipid content of coffee in the Introduction section.

Page 4, line 112-113: "Moreover, it has been shown that the lipid content of boiled unfiltered coffee may be as much as 60 times higher than the lipid content of filtered coffee (29)."

R3-Q6. The authors do not appear to distinguish drinkers from non-drinkers, applying instead a linear model to quantity of consumed coffee/tea. They should provide information about the fraction not drinking at all (apologies if I missed this). If this fraction is minimal, there isn't a problem. However, if a substantial proportion of the study population abstains, then it may be important to consider analysis restricted to drinkers, or comparing drinkers to non-drinkers, e.g. with a logistic model.

Response. We acknowledge the reviewer's comment and appreciate the opportunity to address this issue in the manuscript. We have now performed stratified analysis on coffee consumption frequency in the largest contributing cohort, Framingham Heart Study (FHS), and the Rotterdam Study (RS). To define the groups based on their coffee drinking habits we used a large-scale GWAS on coffee consumption (7), which includes many of the cohorts in our EWAS meta-analysis. Further details of this analysis have been added to the Method section and the results are reported in supplementary Table S9.

a. Methods

Page 8, line 211-215: "In addition, we stratified two of the largest cohorts (FHS and RS) by sex, smoking status and coffee consumption frequency. Smoking status was evaluated on current vs never smokers; coffee consumption frequency was determined as none or infrequent drinkers (<1 cup/day), moderate drinkers (>=1 and <4 cup/day) and high or frequent drinkers (>=4 cup/day) using a large-scale GWAS on coffee consumption as a reference for categorization of the variable (7)."

b. Results

Page 14, line 384-386: “Among the subjects with low coffee consumption, only one CpG remained nominally significant (cg14476101, P=0.03). Likewise, we observed that four of the coffee-associated CpGs remained nominally significant in moderate drinkers.”

R3-Q7. Do the authors think that coffee drinking confounds analysis for tea, and vice versa? Should this be addressed in the analysis or at least in the Discussion? How do the authors account for individuals who drink both coffee and tea? It would not be surprising if there are individuals who drink very little tea but a large amount of coffee. What would the authors expect of their methylation signal compared to individuals who drink a lot of tea? This thread may be relevant to the discussion starting on line 490.

Response. The reviewer’s point is well taken and we thank him/her for having made this comment. Indeed, nutritional epidemiology is a complex field and due to the complexity of human diet, this might also be the case for other beverages (e.g., sugar-sweetened and caffeinated beverages). Fitting the model for some or all of these beverages might introduce overfitting. Prior to our model adjustment in the analysis, we went through the literature and found smoking as the main confounder, which has not been adequately adjusted in prior studies on coffee consumption (30). In addition to smoking, we adjusted the models for age, sex, cell counts, technical covariates, BMI, and alcohol consumption. As suggested by the reviewer, we have now performed sensitivity analyses in the two largest cohorts (FHS and RS), where we have included an additional adjustment for tea in coffee EWAS and vice versa. A description of these sensitivity analyses was added to the Method section and the findings are shown in supplementary Table S9.

a. Methods

Page 7, line 195-197: “Tea and coffee consumption were added as covariates to Model 2 of EWAS on coffee and tea, respectively, in order to assess potential confounding effects on DNA methylation. This analysis was conducted on FHS, the largest cohort, and the RS.”

b. Results

Page 14, line 386-393: “When adjusting by tea consumption, the effect estimates from the majority (eight CpGs) of the identified methylation sites did not change substantially. We observed a similar pattern (same direction in the effect estimate) for the rest of the CpGs that had somewhat change in the effect estimate compared to the main model (Table S9). For example, our main model showed a $\beta = -0.0011$ for cg21161138 ($P = 6.66 \times 10^{-12}$), while the additional adjustment for tea consumption in a subset of cohorts (FHS and RS) showed a smaller effect ($\beta = -0.0003$). The forest plot for this CpG (Figure S5) showed many of the contributing cohorts have a negative effect estimate of the CpG, that are not in our sensitivity analysis. Hence, we postulate that the observed change of the effect estimates could be the result of this, rather than inadequate adjustment for tea consumption.”

R3-Q8. The authors may consider trying to address unique effects of coffee or tea, e.g. by restricting to strata of drinkers exclusively of one or the other.

Response. We acknowledge the reviewer's suggestion. We checked the proportion of exclusive tea drinkers (n=71, 6.6%) and exclusive coffee drinkers (n=299, 28%) in the Rotterdam Study (N total= 1.065). Insight in these numbers suggests that even though we might be able to address the unique effects more thoroughly in terms of statistical analysis, we would lose substantial power to detect any significant association. In particular that in the meta-analysis of overall sample size (~16,000), we have observed relatively modest effect estimates for coffee and tea consumption. Furthermore, given the complexity of human diet itself (e.g., consumption of other beverages like milk and soft drinks), we might not yet reach conclusive results to address the unique effects of coffee or tea.

R3-Q9. Is there any evidence of association, e.g. nominal significance, with tea consumption at loci related to coffee consumption or with coffee consumption at loci (suggestively) related to tea consumption?

Response. We have looked up for the association of coffee-CpGs on the EWAS of tea consumption and vice versa. The results are reported in supplementary Table S11.

Page 15, line 394-396: "Additionally, we observed that three of the coffee-associated CpGs (cg21161138, cg25648203, and cg03636183) showed nominal significant association with tea consumption (Table S11)."

R3-Q10. Could various sensitivity analyses implied by the preceding be performed? The authors could limit this analysis to the largest contributing cohort, the FHS, to reduce workload.

Response. We sincerely appreciate the thoughtful comment of the reviewer as well as understanding the workload. As suggested by the reviewer, we have now performed the recommended sensitivity analyses in the FHS as well as the RS. Please see our responses above (RS1-Q5 and RS3-Q6).

R3-Q11. These considerations are partly related to the issue of whether caffeine or some other component of coffee is driving the associations. It's not critical but the authors could consider looking at caffeine from all sources, i.e. combining tea, coffee, caffeinated beverages, and ask whether the effect they observe are stronger than for coffee alone. This analysis, for example, might be done in just the largest cohort (i.e. the FHS).

Response. Indeed, the evaluation of caffeine alone can be challenging as it comes from the other sources of food as well (e.g., cola, chocolate, energy drinks, caffeine-containing dietary supplement, and even tea in smaller quantity than coffee). Nutrition data in the majority of contributing cohorts were collected by food frequency questionnaires (FFQ). Unfortunately,

in the FFQ, we lack information on some important caffeine sources other than coffee and tea such as cola and energy drinks, and also there is large variations in the other sources of caffeine (such as chocolate) that are available in the FFQ. Therefore, it is hard to estimate caffeine intake from the currently available nutrition data for this study. Moreover, combining these traits may contribute largely to increase the heterogeneity in the association results. It is one of the limitations of our study and we have addressed it in the Discussion.

Page 24, line 646-651: “It is also important to address the potential limitation of this study resulting from the method of dietary data collection via FFQ. While the FFQ are the most cost-effective way to collect the dietary data, they are subject to errors (e.g., participants might have a recall bias when filling in the questionnaires). Nevertheless, when compared to the other dietary assessment methods (e.g. 24-h recall, food diary, etc.), the FFQ usually reports a lower within-person variation”

R3-Q12. It’s not totally clear why model 2 is labeled the “most conservative model.” Including covariates always runs a risk of inducing associations, e.g. by collider bias, even if it may also account for known confounding.

Response. We agree with the reviewer that it is overstating to call the fully adjusted model as the “most conservative”; hence, we have now modified this in the manuscript.

R3-Q13. What is the basis for thinking, a priori, that methylation patterns will be different in EA and AA, justifying stratified analysis (lines 211 & 388)? Are coffee/tea drinking habits very different in the two ancestries? One might expect that the same genes would be involved in both ancestry groups and the sequences around the genes should be essentially identical. Perhaps they could discuss nominal evidence for reciprocal associations, e.g. EA top loci in AA, and vice versa (Table S6)? Can the authors cite evidence for well-powered lack of trans-ancestry replication of methylation sites in other studies? If so, what are the biological mechanisms that result in (validated) methylation differences between EA and AA.

Response. Indeed, drinking coffee and tea are influenced by geographic regions and cultural differences as well as personal preference, which can also be reflected in DNA methylation changes. Evidence has shown that ancestry information signal is mirrored in genome-wide DNA methylation as one of the main variance components in the data (31). Moreover, on the population level, epigenetic profiles appear to differ across ancestral groups (32). For instance, both at birth and in adulthood, African Americans have lower genome-wide levels of methylation compared with individuals of European ancestry (33, 34). A well established example is DNA sequence probed for methylation in the *GATA4* gene which illustrates how differential allele-specific methylation across populations might occur. The SNP rs61277615 in *GATA4* can either be a CpG site with the potential to be methylated when the C allele is present or lack the ability to be methylated when the T allele is present. In populations of European and Asian origin, the C allele has about a 90% frequency, so allele-specific methylation in most individuals is unlikely, while the C and T alleles are equally (each 50%) represented in African population (35).

In light of the evidence, the assessment of DNA methylation in populations with different ancestry, implemented in our research as well as other studies (11, 14, 36), have highlighted the importance of accounting for ancestry information in DNA methylation studies of diverse populations. In our results, of the 9 CpGs significantly associated with coffee consumption in EA participants, none of them replicated in AA participants after adjusting the nominal significant p-value for multiple testing ($P < 0.05/9 = 0.005$). Only two of the CpGs (cg05575921 and cg25648203) had nominal significance p-values in AA subject. Moreover, EWAS in AA participants found one CpG site (cg05822739) associated with coffee consumption ($P = 1.08 \times 10^{-7}$, $\beta = -0.0015$) which was not identified in EA participants, even at the nominal significance level, with much larger sample size. We have now elaborated on this matter in the Discussion as follow:

Page 23, line 618-630: “DNA methylation can differ across continental ancestries (37), challenging replication across populations of varying descent in epigenetic studies (17, 38). Here, our ancestry-stratified EWAS showed lack of similarities of several CpGs in Europeans and African Americans. In particular, none of the coffee-associated CpGs from European populations were replicated in African Americans. Moreover, EWAS in AA participants showed one CpG site (cg05822739) associated with coffee consumption that was not identified in EA participants despite larger sample size. These discrepancies between continental ancestries may be explained by the following reasons: first, some CpG sites may be ancestry-specific, which has to be confirmed by further studies. Second, given the strong influence of culture on coffee drinking habit, its frequency may differ among different ancestry groups. For example, in EA cohorts, the mean coffee consumption intake was 2.3 cups/day, while in AA cohorts this number dropped to 0.79 cups/day. Finally, the difference in sample sizes in our meta-analysis (EA=13,146; AA=2,921) might also have been an important determinant in the discrepancies observed among the populations”

R3-Q14. The forest plots from each significant locus do not show the overall estimate. Please provide. It’s also clear from the presentation that the effects are dominated by the FHS contribution. Would the authors consider sensitivity analysis that removes the FHS contribution, or examines the FHS association alone?

Response. We have adapted the forest plots as suggested. Moreover, we have excluded FHS from the meta-analysis and have reported the findings in the Results section accordingly.

Page 13, line 346-349: “When excluding the largest contributing cohort (FHS) from the meta-analysis of European cohorts, four CpGs namely cg05575921 ($P = 1.2 \times 10^{-14}$, $\beta = -0.003$), cg25648203 ($P = 5.4 \times 10^{-10}$, $\beta = -0.001$), cg21161138 ($P = 5.7 \times 10^{-10}$, $\beta = -0.001$) and cg03636183 ($P = 4.1 \times 10^{-8}$, $\beta = -0.001$) remained significant.”

R3-Q15. Table 2 should show standard errors for the beta estimates.

Response. We have added standard errors to Table 2 in the revised manuscript.

R3-Q16. The *AHRR* gene is a great candidate for effects of coffee. However, the authors note that the top CpGs are eQTM for *AHRR* but for the neighboring gene *EXOC3*. Do they think there may be an eQTM for *AHRR* in a different tissue, e.g. liver, as is apparently demonstrated for the CpG near *PHGDH* in ref. 54.

Response. Due to tissue-specific expression patterns of genes, it is indeed possible to see an eQTM for *AHRR* gene in a different tissue (e.g. liver). We checked different resources, specifically described by Walton et al (39); however, none of these sources provide specific information of eQTMs in other tissues, different than blood.

Page 19, line 500-503: “The four CpGs annotated to *AHRR* showed an association with expression of the neighbouring gene *EXOC3* in blood. However, due to tissue-specific expression patterns of genes, it is also possible to see eQTM for *AHRR* gene in a different tissue (e.g. liver) that warrant investigation in future studies.”

R3-Q17. For a null MR results, such as here, one typically would not perform sensitivity analysis as the authors did. It’s not totally clear how a significant result in the sensitivity analysis would be reconciled with null primary analysis. Yes, there could be an argument that primary effects are confounded toward the null due to pleiotropy. However, if the methods that try to account for pleiotropy were to yield a significant result, there may still remain questions about whether the finding may still be driven by pleiotropy that is not addressed by the sensitivity methods. In the MR, there is also no discussion of how genetic effects on methylation may vary by tissue, even that the cis-meQTL data derives from blood and other tissues may be relevant, e.g. liver (Methods, and Discussion starting line 465).

Response. We appreciate the reviewer for raising this remark and agree on the suggestion of not reporting the MR sensitivity analyses when IVW results are null. Therefore, we have adapted the corresponding Table keeping Egger and weighted median estimates when IVW effect is significant.

Regarding meQTLs for other tissues, different than blood, we have explored resources like GTEx (<https://www.gtexportal.org/>); unfortunately, they have not reported meQTLs measured in liver. As suggested by the reviewer, we have now elaborated further on this point in the Discussion.

Page 21, line 562-564: “As the analyses have been explored using meQTL data obtained from blood, we cannot rule out the potential differences that might be observed in other relevant tissues (e.g., liver).”

R3-Q18. In spite of the authors claims to the contrary, causality in the conclusions is in doubt (p. 18). The MR was negative but this result is attributed to lack of power, even in the absence of a power calculation, which should be provided. The authors cite additional reasons for a null MR result. However, not mentioned is the alternative conclusion that the methylation patterns reflect pathways of coffee consumption effects on the body but are not

causal. It's not completely clear what the arrows imply in figure S2. However, to the extent they represent causality, the figure is incomplete in not considering reverse directions, e.g. from liver function to methylation, or from coffee consumption to methylation, or even from expression to methylation, the latter, e.g. in trans.

Response. We thank the reviewer for this comment. Regarding power calculation of the MR studies, a post-hoc power calculation may not offer a clear representation of the sample size needed to obtain a significant causality effect; especially when the DNA methylation sample is not large enough, which is evidenced in small effect sizes and R^2 (variance explained) values. In addition, we consider that the power estimate may be affected by horizontal pleiotropy, which is not accounted for by the power calculation tools available. In line with our reasoning, the majority of previous EWAS studies implementing the two-sample MR approach do not provide a specific description of power calculation (40, 41). Evidence has shown that it is better to interpret the confidence intervals, which we have now added to our manuscript. Moreover, it is worth to mention that a recent review of MR studies of coffee and caffeine consumption indicates that substantial sample size is needed to assess the causality (22). The review shows that the majority of the studies reported non significant findings and stated underpowered IV as a limitation despite substantial sample sizes (in some cases $N \geq 100,000$). Therefore, extensive sample sizes might be needed in order to provide substantial power and draw causal inference on coffee consumption. Indeed, one alternative conclusion could also be that the methylation patterns reflect pathways of coffee consumption effects on the body but are not causal. We have now incorporated this in the Discussion. Moreover, we have adapted Figure S2 as kindly suggested by the reviewer.

Page 20, line 542-552: “We attempted to provide additional evidence of directionality of effect of coffee consumption on cg14476101 using MR methods; however, the power of our MR analyses was limited apparently due to the small sample size of our DNA methylation data, which is evident in small effect sizes and R^2 (variance explained) values. The lack of strong genetic instruments for coffee exposure might also be an important determinant of the results observed. In line with this reasoning, a recent review of MR studies of coffee and caffeine consumption (22) has indicated that substantial sample size is indeed needed to assess the causality when it comes to coffee and other dietary habits. The majority of the studies included in this review reported non-significant findings and stated underpowered instrumental variables as a limitation despite substantial sample sizes (in some cases $N \geq 100,000$). Therefore, for MR studies on coffee consumption, extensive sample sizes might be needed in order to provide substantial power and infer causality”

Page 21, line 560-563: “An alternative conclusion could also be that the methylation patterns reflect pathways of coffee consumption effects on the body but they not necessarily causal.”

R3-Q19. The comments about smoking are interesting. The authors note that smoking is associated with methylation at some of the same sites. They also note that smoking and coffee drinking are correlated, but even after adjusting for smoking status, the associations between coffee consumption and methylation at these sites with coffee consumption persists. Are the effects at these sites associated with smoking in the same direction as the effects associated with coffee drinking? Please comment/address.

Response. We have curated information from EWAS catalog (<http://www.ewascatalog.org/>), EWAS atlas (<https://bigd.big.ac.cn/ewas>) and Google scholar in order to check the direction of effect of the coffee-associated CpGs in the association with smoking. We have provided these information in the Results section (supplementary Table S10). Data was not found for four coffee-associated CpGs namely: cg14476101, cg06690548, cg15928106 and cg20228731, but these are not located in ‘smoking genes’.

a. Results:

Page 14, line 377-383: “As the smoker-status covariate in our study was discrete (never, former or current smoker), the adjusted EWAS model might have been not adequately controlled for smoking exposure (e.g., duration of smoking, amount of smoking for a period of time, and second hand smoking). Therefore, we looked up the coffee-associated CpGs in the published EWAS on smoking behaviour (Table S10). Eight of the lead coffee-associated CpGs have been previously linked to smoking. This may suggest that residual smoking exposure remains potential confounding factor for the probe association with coffee consumption.”

b. Discussion:

Page 24, line 635-638: “One important concern regarding this analysis is smoking, given that the effect of smoking on DNA methylation has been recognized (42) and previous studies have shown that heavier smokers tend to drink more coffee (43). Even though we adjusted for smoking in our analysis, there is likely to be some residual confounding of smoking”

R3-Q20. Please refine the discussion of HDAC4. The authors note its role in xenobiotic metabolism but its molecular function is as a histone deacetylase. Is xenobiotic metabolism the only pathway in which it is active?

Response. We thank the reviewer for this comment and have now refined the Discussion accordingly.

Discussion:

Page 19, line 514-518: “The second gene, *HDAC4* encodes histone deacetylase with a molecular role of deacetylation of lysine residues of the core histones (44), where histone modifications are another epigenetic mechanism. This might be an indication of potential interplay between two epigenetic modifications. Moreover, *HDAC4* gene is a member of the Xenobiotic metabolism signalling”.

Page 20, line 523-525: “It is interesting to note that *HDAC4* and *HDAC5* belong to the same *HDAC* classification (44), and *HDAC5* gene has been associated previously with cocaine dependence in a recent EWAS of cocaine and crack dependents (20).”

R3-Q21. How do the authors justify genome-wide threshold in this experiment? Is there a reference for the threshold of $p < 10^{-7}$ in this kind of methylation scan?

Response. The majority of the cohorts included in this meta-analysis have used the Illumina 450k Infinium methylation beadchip, which interrogates ~450,000 methylation sites (CpGs) per sample at single nucleotide resolution. In order to counteract the multiple comparison problem (because of testing 450K CpGs), we have used a Bonferroni-adjusted threshold of $\alpha = 10^{-7}$. This significance threshold of p-value $< (0.05/450,000) 1.1 \times 10^{-7}$ is widely accepted and recognized in this kind of epigenome-wide DNA methylation scan (11-14). For more information please see our response to R1-Q9.

References

1. Matoba N, Akiyama M, Ishigaki K, Kanai M, Takahashi A, Momozawa Y, et al. GWAS of 165,084 Japanese individuals identified nine loci associated with dietary habits. *Nature human behaviour*. 2020;4(3):308-16.
2. Seale P, Conroe HM, Estall J, Kajimura S, Frontini A, Ishibashi J, et al. Prdm16 determines the thermogenic program of subcutaneous white adipose tissue in mice. *The Journal of clinical investigation*. 2011;121(1):96-105.
3. Zhang JH, Li NF, Yan ZT, Wang HM, Guo YY, Ling Z. Association of genetic variations of PRDM16 with metabolic syndrome in a general Xinjiang Uygur population. *Endocrine*. 2012;41(3):539-41.
4. Li NF, Wang HM, Bi YW, Zhou L, Yao XG, Yan ZT, et al. Association study of PRDM16 gene polymorphisms with essential hypertension in Xinjiang Uygur population. *Zhonghua yi xue yi Chuan xue za zhi= Zhonghua Yixue Yichuanxue Zazhi= Chinese Journal of Medical Genetics*. 2013;30(6):716-20.
5. Chen Y-a, Lemire M, Choufani S, Butcher DT, Grafodatskaya D, Zanke BW, et al. Discovery of cross-reactive probes and polymorphic CpGs in the Illumina Infinium HumanMethylation450 microarray. *Epigenetics*. 2013;8(2):203-9.
6. Watanabe K, Stringer S, Frei O, Mirkov MU, de Leeuw C, Polderman TJC, et al. A global overview of pleiotropy and genetic architecture in complex traits. *Nature genetics*. 2019;51(9):1339-48.
7. Coffee, Caffeine Genetics C, Cornelis MC, Byrne EM, Esko T, Nalls MA, et al. Genome-wide meta-analysis identifies six novel loci associated with habitual coffee consumption. *Mol Psychiatry*. 2015;20(5):647-56.
8. Thorn CF, Aklillu E, McDonagh EM, Klein TE, Altman RB. PharmGKB summary: caffeine pathway. *Pharmacogenetics and genomics*. 2012;22(5):389.
9. Amin N, Byrne E, Johnson J, Chenevix-Trench G, Walter S, Nolte IM, et al. Genome-wide association analysis of coffee drinking suggests association with CYP1A1/CYP1A2 and NRCAM. *Molecular psychiatry*. 2012;17(11):1116-29.
10. Ek WE, Tobi EW, Ahsan M, Lampa E, Ponzi E, Kyrtopoulos SA, et al. Tea and coffee consumption in relation to DNA methylation in four European cohorts. *Human molecular genetics*. 2017;26(16):3221-31.
11. Ligthart S, Marzi C, Aslibekyan S, Mendelson MM, Conneely KN, Tanaka T, et al. DNA methylation signatures of chronic low-grade inflammation are associated with complex diseases. *Genome biology*. 2016;17(1):1-15.
12. Linnér RK, Marioni RE, Rietveld CA, Simpkin AJ, Davies NM, Watanabe K, et al. An epigenome-wide association study meta-analysis of educational attainment. *Molecular psychiatry*. 2017;22(12):1680-90.
13. Küpers LK, Monnereau C, Sharp GC, Yousefi P, Salas LA, Ghantous A, et al. Meta-analysis of epigenome-wide association studies in neonates reveals widespread differential DNA methylation associated with birthweight. *Nature communications*. 2019;10(1):1-11.
14. Chu AY, Tin A, Schlosser P, Ko Y-A, Qiu C, Yao C, et al. Epigenome-wide association studies identify DNA methylation associated with kidney function. *Nature communications*. 2017;8(1):1-12.
15. Nardone S, Sams DS, Reuveni E, Getselter D, Oron O, Karpuj M, et al. DNA methylation analysis of the autistic brain reveals multiple dysregulated biological pathways. *Translational psychiatry*. 2014;4(9):e433-e.
16. Grundberg E, Meduri E, Sandling JK, Hedman ÅK, Keildson S, Buil A, et al. Global analysis of DNA methylation variation in adipose tissue from twins reveals links to disease-associated variants in distal regulatory elements. *The American Journal of Human Genetics*. 2013;93(5):876-90.
17. Liu C, Marioni RE, Hedman ÅK, Pfeiffer L, Tsai P-C, Reynolds LM, et al. A DNA methylation biomarker of alcohol consumption. *Molecular psychiatry*. 2018;23(2):422-33.
18. Joehanes R, Just AC, Marioni RE, Pilling LC, Reynolds LM, Mandaviya PR, et al. Epigenetic signatures of cigarette smoking. *Circulation: cardiovascular genetics*. 2016;9(5):436-47.
19. Dhana K, Braun KVE, Nano J, Voortman T, Demerath EW, Guan W, et al. An epigenome-wide association study of obesity-related traits. *American journal of epidemiology*. 2018;187(8):1662-9.
20. Camilo C, Maschietto M, Vieira HC, Tahira AC, Gouveia GR, Feio dos Santos AC, et al. Genome-wide DNA methylation profile in the peripheral blood of cocaine and crack dependents. *Brazilian Journal of Psychiatry*. 2019;41(6):485-93.
21. Osborne AJ, Pearson JF, Noble AJ, Gemmell NJ, Horwood LJ, Boden JM, et al. Genome-wide DNA methylation analysis of heavy cannabis exposure in a New Zealand longitudinal cohort. *Translational Psychiatry*. 2020;10(1):1-10.
22. Cornelis MC, Munafo MR. Mendelian randomization studies of coffee and caffeine consumption. *Nutrients*. 2018;10(10):1343.

23. Rein MJ, Renouf M, Cruz-Hernandez C, Actis-Goretta L, Thakkar SK, da Silva Pinto M. Bioavailability of bioactive food compounds: a challenging journey to bioefficacy. *British journal of clinical pharmacology*. 2013;75(3):588-602.
24. Juvinao-Quintero DL, Hivert M-F, Sharp GC, Relton CL, Elliott HR. DNA Methylation and Type 2 Diabetes: the Use of Mendelian Randomization to Assess Causality. *Current genetic medicine reports*. 2019;7(4):191-207.
25. Sayols-Baixeras S, Tiwari HK, Aslibekyan SW, editors. Disentangling associations between DNA methylation and blood lipids: a Mendelian randomization approach. *BMC proceedings*; 2018: Springer.
26. Richardson TG, Zheng J, Smith GD, Timpson NJ, Gaunt TR, Relton CL, et al. Mendelian randomization analysis identifies CpG sites as putative mediators for genetic influences on cardiovascular disease risk. *The American Journal of Human Genetics*. 2017;101(4):590-602.
27. Bennett DA, Landry D, Little J, Minelli C. Systematic review of statistical approaches to quantify, or correct for, measurement error in a continuous exposure in nutritional epidemiology. *BMC medical research methodology*. 2017;17(1):146.
28. Landais E, Moskal A, Mullee A, Nicolas G, Gunter MJ, Huybrechts I, et al. Coffee and tea consumption and the contribution of their added ingredients to total energy and nutrient intakes in 10 European countries: Benchmark data from the late 1990s. *Nutrients*. 2018;10(6):725.
29. Gross G, Jaccaud E, Huggett AC. Analysis of the content of the diterpenes cafestol and kahweol in coffee brews. *Food and Chemical Toxicology*. 1997;35(6):547-54.
30. Thomas DR, Hodges ID. Dietary research on coffee: Improving adjustment for confounding. *Current developments in nutrition*. 2020;4(1):nzz142.
31. Rahmani E, Shenhav L, Schweiger R, Yousefi P, Huen K, Eskenazi B, et al. Genome-wide methylation data mirror ancestry information. *Epigenetics & chromatin*. 2017;10(1):1-12.
32. Fraser HB, Lam LL, Neumann SM, Kobor MS. Population-specificity of human DNA methylation. *Genome biology*. 2012;13(2):1-12.
33. Adkins RM, Krushkal J, Tylavsky FA, Thomas F. Racial differences in gene-specific DNA methylation levels are present at birth. *Birth Defects Research Part A: Clinical and Molecular Teratology*. 2011;91(8):728-36.
34. Zhang FF, Cardarelli R, Carroll J, Fulda KG, Kaur M, Gonzalez K, et al. Significant differences in global genomic DNA methylation by gender and race/ethnicity in peripheral blood. *Epigenetics*. 2011;6(5):623-9.
35. Schalkwyk LC, Meaburn EL, Smith R, Dempster EL, Jeffries AR, Davies MN, et al. Allelic skewing of DNA methylation is widespread across the genome. *The American Journal of Human Genetics*. 2010;86(2):196-212.
36. Husquin LT, Rotival M, Fagny M, Quach H, Zidane N, McEwen LM, et al. Exploring the genetic basis of human population differences in DNA methylation and their causal impact on immune gene regulation. *Genome biology*. 2018;19(1):1-17.
37. Barfield RT, Almlí LM, Kilaru V, Smith AK, Mercer KB, Duncan R, et al. Accounting for population stratification in DNA methylation studies. *Genetic epidemiology*. 2014;38(3):231-41.
38. Ma J, Rebholz CM, Braun KVE, Reynolds LM, Aslibekyan S, Xia R, et al. Whole Blood DNA Methylation Signatures of Diet Are Associated with Cardiovascular Disease Risk Factors and All-cause Mortality. *Circulation: Genomic and Precision Medicine*. 2020.
39. Walton E, Relton CL, Caramaschi D. Using Openly Accessible Resources to Strengthen Causal Inference in Epigenetic Epidemiology of Neurodevelopment and Mental Health. *Genes*. 2019;10(3):193.
40. Kwok MK, Leung GM, Schooling CM. Habitual coffee consumption and risk of type 2 diabetes, ischemic heart disease, depression and Alzheimer's disease: a Mendelian randomization study. *Scientific reports*. 2016;6:36500.
41. Kennedy OJ, Pirastu N, Poole R, Fallowfield JA, Hayes PC, Grzeszkowiak EJ, et al. Coffee Consumption and Kidney Function: A Mendelian Randomization Study. *American Journal of Kidney Diseases*. 2019.
42. Joehanes R, Just AC, Marioni RE, Pilling LC, Reynolds LM, Mandaviya PR, et al. Epigenetic Signatures of Cigarette Smoking. *Circ Cardiovasc Genet*. 2016;9(5):436-47.
43. Bjorngaard JH, Nordestgaard AT, Taylor AE, Treur JL, Gabrielsen ME, Munafo MR, et al. Heavier smoking increases coffee consumption: findings from a Mendelian randomization analysis. *Int J Epidemiol*. 2017;46(6):1958-67.
44. Seto E, Yoshida M. Erasers of histone acetylation: the histone deacetylase enzymes. *Cold Spring Harbor perspectives in biology*. 2014;6(4):a018713.

REVIEWERS' COMMENTS

Reviewer #1 (Remarks to the Author):

The authors have answered and implemented all my suggestions and questions. I feel satisfied

Reviewer #2 (Remarks to the Author):

Authors fully responded the reviewer's comments, including additional assessments of the tea consumption EWAS. This reviewer does not have further comments.

Reviewer #3 (Remarks to the Author):

The manuscript "Epigenome-wide association meta-analysis of DNA methylation with coffee and tea consumption" by Karabegovic et al. has been thoughtfully revised in response to the reviewers' comments. In particular, the stratification of the analysis by smoking status addresses reviewers' concerns while also strengthening the investigators' findings. This is a very substantial study.

Comments (in order of appearance)

1. A remaining technical issue is whether tea associations should be described as suggestive or null. Technically, the latter is more accurate. However, the authors prefer to interpret the suggestive associations, perhaps introducing more speculation than is wise. Please consider removing the biological interpretation of these loci or abbreviating (Abstract, Results, Discussion).

2. Lines 115-122 Please streamline this section. There are repeated thoughts.

3. Line 336 statement is a little difficult to understand: "not to miss any potential signals due to lack of power"

4. Line 443-471 The investigation into PHGDH is interesting and adds to the manuscript. However, it begs the question of whether genetic variation near this gene is associated in GWAS with fatty liver disease. The GWAS catalog suggests it is not in spite of GWAS associations with red blood cell properties, metabolites, and cholesterol. Is it possible there is a TWAS signal at this gene with fatty liver disease? Do the authors have an explanation for the apparent lack of a genetic signal with some phenotypes but not fatty liver? Does this situation temper their conclusions?

5. The Discussion make excellent points, some in response to the reviewers' comments, but is also very (i.e. too) long and unwieldy, again in part due to a conscientious effort by the authors to respond to the reviewers. It could be streamlined with minimal loss of impact. Editing for length could include, but should not be limited to, the following:

-- para beginning on line 578 could be condensed

-- para beginning on line 596 is probably too speculative given that the tea associations are only suggestive. Recommend removing.

-- para beginning on line 606 doesn't add much as long as the lack of associations at the coffee candidate genes is noted in the Results

-- para beginning on line 618 could be condensed

-- line 630 in this paragraph should be the start of a new para on strengths and limitations.

-- last para beginning on line 654 repeats ideas that have already been stated and could be condensed to a 1 or 2 sentence conclusion.

Minor

There are missing articles, i.e. "a" and/or "the" in some places. Please review.

REVIEWER COMMENTS

Reviewer #3 (Remarks to the Author):

The manuscript “Epigenome-wide association meta-analysis of DNA methylation with coffee and tea consumption” by Karabegovic et al. has been thoughtfully revised in response to the reviewers’ comments. In particular, the stratification of the analysis by smoking status addresses reviewers’ concerns while also strengthening the investigators’ findings. This is a very substantial study.

1. A remaining technical issue is whether tea associations should be described as suggestive or null. Technically, the latter is more accurate. However, the authors prefer to interpret the suggestive associations, perhaps introducing more speculation than is wise. Please consider removing the biological interpretation of these loci or abbreviating (Abstract, Results, Discussion).

Response. We would like to thank the reviewer for the positive and constructive comments. Regarding the two suggestive loci for tea consumption, we have now modified the Abstract and Results accordingly and removed the biological interpretation of these tea loci from Discussion.

It is worth mentioning that the epigenome-wide significant threshold of 1.1×10^{-7} is based on the stringent Bonferroni correction for the number of tested CpGs ($0.05/450,000 = 1.1 \times 10^{-7}$), which is widely used in the previous EWASs. However, it has been shown that there are co-methylation regions (DMRs) in the illumina 450k methylation array. Hence some previous EWASs have suggested FDR-based thresholds of 1% to 5% as the significance thresholds, which correspond to p-values close to 1×10^{-4} . In our study, we considered a suggestive threshold of 5.0×10^{-6} between these thresholds recommended by previous studies, in order to detect also suggestive signals and not to miss any important loci because of a lack of power. We thought the best way of presenting the tea suggestive association is to describe them as null, as suggested by the reviewer, mentioning that they have not passed the EWAS significant threshold, but report the two CpGs most significantly associated with tea consumption in the Results (without biological interpretation of the loci in Discussion). This is fairly common in the EWAS/GWAS field that researchers report non-significant, but borderline associations that are just below the significant threshold. This could provide an opportunity for future studies to replicate and meta-analysis these suggestive signals with their data, which then may surpass the EWAS significant threshold.

2. Lines 115-122 Please streamline this section. There are repeated thoughts.

Response. We thank the reviewer for pointing this out. We have modified the text accordingly.

3. Line 336 statement is a little difficult to understand: “not to miss any potential signals due to lack of power”

Response. We agree with the reviewer and have removed the statement.

4. Line 443-471 The investigation into PHGDH is interesting and adds to the manuscript. However, it begs the question of whether genetic variation near this gene is associated in GWAS with fatty liver disease. The GWAS catalog suggests it is not in spite of GWAS associations with red blood cell properties, metabolites, and cholesterol. Is it possible there is a TWAS signal at this gene with fatty liver disease? Do the authors have an explanation for the apparent lack of a genetic signal with some phenotypes but not fatty liver? Does this situation temper their conclusions?

Response. Indeed, no common genetic variation near *PHGDH* has been reported so far to be associated with NAFLD in the previous GWAS. However, an intergenic variant (rs454510) between *PHGDH* and *ZNF697* genes has been shown to be associated with alcohol-related liver cirrhosis [PMID: 28714907] or an intronic variant within *PHGDH* has been linked to Alanine aminotransferase (ALT) levels [PMID: 28090653]. Moreover, as also mentioned by the reviewer, SNPs in *PHGDH* have been associated with serum metabolites levels [PMID: 24816252] and cholesterol [PMID: 30275531], which is in line with the results of our experiments indicating a potential role of this gene in hepatic-lipid metabolisms. Maybe future large-scale GWAS of NAFLD, including also uncommon and rare variants, could show the association with NAFLD. To the best of our knowledge there are no TWAS signals in this gene previously reported for liver diseases. Yet previous evidence has indicated that a reduced expression of *PHGDH* is linked to the development of fatty liver disease [PMID: 31678070]. We have added the information about association between *PHGDH* genetic variants and cholesterol levels to the discussion.

5. The Discussion make excellent points, some in response to the reviewers' comments, but is also very (i.e. too) long and unwieldy, again in part due to a conscientious effort by the authors to respond to the reviewers. It could be streamlined with minimal loss of impact. Editing for length could include, but should not be limited to, the following:

- para beginning on line 578 could be condensed
- para beginning on line 596 is probably too speculative given that the tea associations are only suggestive. Recommend removing.
- para beginning on line 606 doesn't add much as long as the lack of associations at the coffee candidate genes is noted in the Results
- para beginning on line 618 could be condensed
- line 630 in this paragraph should be the start of a new para on strengths and limitations.
- last para beginning on line 654 repeats ideas that have already been stated and could be condensed to a 1 or 2 sentence conclusion.

Response. We acknowledge the reviewer's comment regarding the lengthy Discussion and appreciate him/her for understanding our effort to respond to all the reviewer's comments and suggestions. As advised by the reviewer, we have now removed the biological interpretation of the tea suggestive loci and also condensed the other mentioned paragraphs, taking into account the impact and flow of the Discussion. The length of Discussion was 2660 words (190 lines), which is now reduced to ~2120 words (151 lines).

Minor

There are missing articles, i.e. "a" and/or "the" in some places. Please review.

Response. We thank the reviewer for noticing this. We have read the manuscript carefully and tried to incorporate "a" or "the" wherever needed through the manuscript.